# An electron-deficient carbon current collector for anode-free Li-metal batteries

Hyeokjin Kwon[1], Ju-Hyuk Lee[1], Youngil Roh[1], Jaewon Baek[1], Dong Jae Shin[1], Jong Keon Yoon[2], Hoe Jin Ha[2], Je Young Kim[2] & Hee-Tak Kim [1,3✉]

The long-term cycling of anode-free Li-metal cells (i.e., cells where the negative electrode is in situ formed by electrodeposition on an electronically conductive matrix of lithium sourced from the positive electrode) using a liquid electrolyte is affected by the formation of an inhomogeneous solid electrolyte interphase (SEI) on the current collector and irregular Li deposition. To circumvent these issues, we report an atomically defective carbon current collector where multivacancy defects induce homogeneous SEI formation on the current collector and uniform Li nucleation and growth to obtain a dense Li morphology. Via simulations and experimental measurements and analyses, we demonstrate the beneficial effect of electron deficiency on the Li hosting behavior of the carbon current collector. Furthermore, we report the results of testing anode-free coin cells comprising a multivacancy defective carbon current collector, a $Li_xNi_{0.8}Co_{0.1}Mn_{0.1}$-based cathode and a nonaqueous Li-containing electrolyte solution. These cells retain 90% of their initial capacity for over 50 cycles under lean electrolyte conditions.

[1] Department of Chemical and Biomolecular Engineering, KAIST, 291 Daehak-ro, Yuseong-gu, Daejeon 34141, Republic of Korea. [2] Battery R&D, LG Energy Solution, 188, Munji-ro, Yuseong-gu, Daejeon 34122, Republic of Korea. [3] Advanced Battery Center, KAIST Institute for the NanoCentury, KAIST, 291, Daehak-ro, Yuseong-gu, Daejeon 34141, Republic of Korea. ✉email: heetak.kim@kaist.ac.kr

To achieve high energy density lithium (Li)-metal batteries, an appropriate negative to positive capacity ratio (N/P < 3), a low electrolyte amount to capacity ratio (E/C < 10 μl mAh$^{-1}$), and a positive electrode with high mass loading (>4 mAh cm$^{-2}$) are considered to be requisite conditions. In this aspect, the anode-free Li-metal battery, which is free of Li metal when initially assembled, represents one of the most advanced concepts of Li-metal cell technology[1–3]. During the first charging of an anode-free cell, Li metal is electroplated on the anode current collector. However, the Li/current collector interface in anode-free cells presents critical challenges, which include cathodic Li corrosion[4] and ramified Li electrodeposition on the current collector[5]. Similar to other types of Li-based cells, the uncontrolled reaction between the roughened Li deposit and electrolyte critically limits the long-term cycling of anode-free cells. This failure of the negative electrode becomes severe at a high cathode loading and low E/C ratio, hindering the improvement in energy density. Considerable efforts have recently been made to control Li plating/stripping on the current collector, including the use of various host materials[6–10] and a coherent lattice[11]. While the efficacy of three-dimensional (3D) porous current collectors in reducing the effective current density has been proven, their enlarged electroactive surface area can accompany a larger degree of electrolyte decomposition at the current collector–electrolyte interface than typical copper current collectors[12].

Regarding the cycling life of anode-free cells, the initial structure and properties of the SEI formed on the current collector are highly important because they influence the sequential evolution of the Li deposit morphology in regard to electrochemical and mechanical aspects[13–15]. The properties of the mosaic and grain-coalesced SEI formed on the current collector surface, which originate from electron transfer from the current collector to the electrolyte, can induce continuous electrolyte decomposition and a defective SEI, thereby promoting uneven initial Li nucleation and growth. In this regard, we envisioned that it is possible to modify the microstructure of the SEI and the behavior of Li nucleation/growth by tuning the Fermi level of the current collector surface. Considering the recently reported cathodic corrosion of Li due to electron transfer from the current collector to the electrolyte[4], a properly tuned electronic structure is of significant interest in terminating interfacial electron transfer. Furthermore, a low Fermi level surface can provide a means to control the Li nucleation and growth mode on the current collector based on the electronic interaction between electropositive Li and the electron-deficient current collector.

In this work, with the concept of tuning the electronic structure of the current collector, we present a multivacancy (MV) defect-enriched carbon surface as the anode current collector of an anode-free cell. In contrast to the formation of a nonuniform SEI and growth of dendritic Li on the graphite surface, the MV defect structure addresses the critical issues at the anode interface by (1) constructing a thin and uniform SEI with reduced electrolyte decomposition and (2) evenly distributing a large number of Li nuclei that laterally grow on the current collector surface. The interfacial compatibility of the MV defect for the anode-free cell originates from its low Fermi level property, which can suppress electron transfer to the lowest unoccupied molecular orbital (LUMO) of the electrolyte and impart the strong binding of Li adatoms via orbital hybridization (electron transfer) (Fig. 1). We report the successful introduction of an MV-defective current collector in an NCM-based anode-free Li metal cell, showing a capacity retention of 90% after 50 cycles at a current density and cathode-limited areal capacity of 2.0 mA cm$^{-2}$ and 4.2 mAh cm$^{-2}$, respectively, and under lean carbonate electrolyte conditions (E/C ratio of 4.0 μl mAh$^{-1}$). We believe that this research work could point toward an atomic and electronic design direction for the current collector to achieve long-lasting anode-free Li-based cells.

## Results

**MV-defective carbon current collector**. We chose a carbon layer with MV defects as an electron-deficient current collector. The defect structure was formed on the graphitic surface of commercial carbon paper by the coating and subsequent carbonization of a zeolitic imidazole framework (ZIF-8). ZIF-8 can be converted to a highly defective carbon structure by heating due to the evaporation of Zn and N atoms[16]. Pristine carbon paper (p-CP) and defective carbon layer-coated carbon paper (d-CP) enable a focused comparison of graphitic and defective surfaces, excluding the effect of the geometry of the current collector.

The structures of p-CP, ZIF-8-coated p-CP (ZIF-8@p-CP), and d-CP were investigated by using scanning electron microscopy (SEM), X-ray photoelectron spectroscopy (XPS), and energy-dispersive X-ray (EDX) spectroscopy mapping. As shown in Fig. 2a, b, the diameter of the carbon fiber was approximately 6.0 μm, and the scale of the interstitial space between adjacent fibers was tens of microns. The SEM images of ZIF-8@p-CP (Supplementary Fig. 1a) show that crystalline ZIF-8 particles covered the carbon fibers of p-CP. The Zn, N, and O signals in the XPS survey spectra and EDX mapping of p-CP, ZIF-8@p-CP, and d-CP confirm the surface decoration of ZIF-8 and the carbonization of ZIF-8 (Supplementary Fig. 1b–f and Supplementary Table 1). The XPS survey spectrum of d-CP shows a very low N content of 1.2% for d-CP with no Zn signal (Supplementary Fig. 1c), indicating the removal of N and Zn from ZIF by the carbonization process[17,18]. During the carbonization process, the locally grown ZIF-8 nanocrystals laterally spread along the carbon fiber surfaces due to the structural collapse of ZIF-8[19], resulting in a webby carbon layer on the surface, as shown in Fig. 2c, d.

High-resolution transmission electron microscopy (HR-TEM) analysis with corresponding fast Fourier transform (FFT) results revealed the difference in the basal plane crystallinity between p-CP and d-CP (Fig. 2e, f). The HR-TEM images of p-CP featuring hexagonally packed carbon atoms with a nearest-neighbor hexagon center separation of 0.246 nm verified its highly crystalline structure[20]. The corresponding FFT analysis revealed a hexagonal-shaped reciprocal-space pattern, demonstrating an $sp^2$-hybridized graphene surface. In contrast, d-CP showed a disordered basal plane structure in its HR-TEM image together with a diffusion halo in its FFT image, which is characteristic of a disordered structure. The bonding structures of the p-CP and d-CP surfaces were quantitatively analyzed by XPS (Fig. 2g). A pronounced difference between the two electrodes was that the peak at 283.7 eV in the C 1s spectra, corresponding to a vacancy defect[21,22], was much larger for d-CP. The peak intensity ratio of vacancies to $sp^2$-hybridized carbon was 0.175 for p-CP and 0.403 for d-CP. The defective structure of d-CP was also confirmed by the Raman spectra of p-CP and d-CP (Fig. 2h). The ratio of the D (1330–1340 cm$^{-1}$) and G (1590–1600 cm$^{-1}$) bands ($I_D/I_G$), which is indicative of defect density, was 1.240 for p-CP and 1.744 for d-CP, indicating the vacancy-rich defective carbon surface of d-CP. The ratio of the 2D band at ~2700 cm$^{-1}$ and the G-band was lower than unity (0.496 for p-CP and 0.603 for d-CP), reflecting the formation of defects on the layered structure[17,18].

The pore structure and size distribution of the defective layer were investigated via nitrogen adsorption-desorption measurements and analyzed by applying the Brunauer–Emmett–Teller (BET) and Barrett–Joyner–Halenda (BJH) theories (Fig. 2i and Supplementary Fig. 2a–d). The isotherm of d-CP featuring a concave quantity-adsorbed curve in the low relative pressure

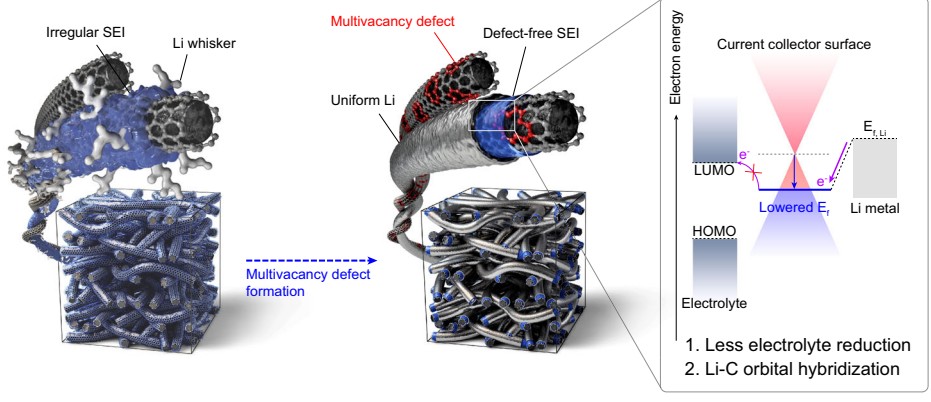

**Fig. 1 Schematic of the dual functionality of an electron-deficient carbon surface.** Pristine carbon fiber electrode with a graphitic surface (p-CP) forming a nonuniform SEI (blue layer) and dendritic Li deposit (silver color protrusion), and defective carbon layer-decorated carbon fiber electrode (d-CP) inducing a defect-free SEI and dendrite-free Li morphology. Suppression of electron transfer to the electrolyte on the defective carbon surface of d-CP via a lower Fermi level (larger work function) in contrast to the graphitic carbon surface of p-CP. Orbital hybridization between the lithium deposits and carbon defects on the defective carbon surface of d-CP due to its low Fermi level.

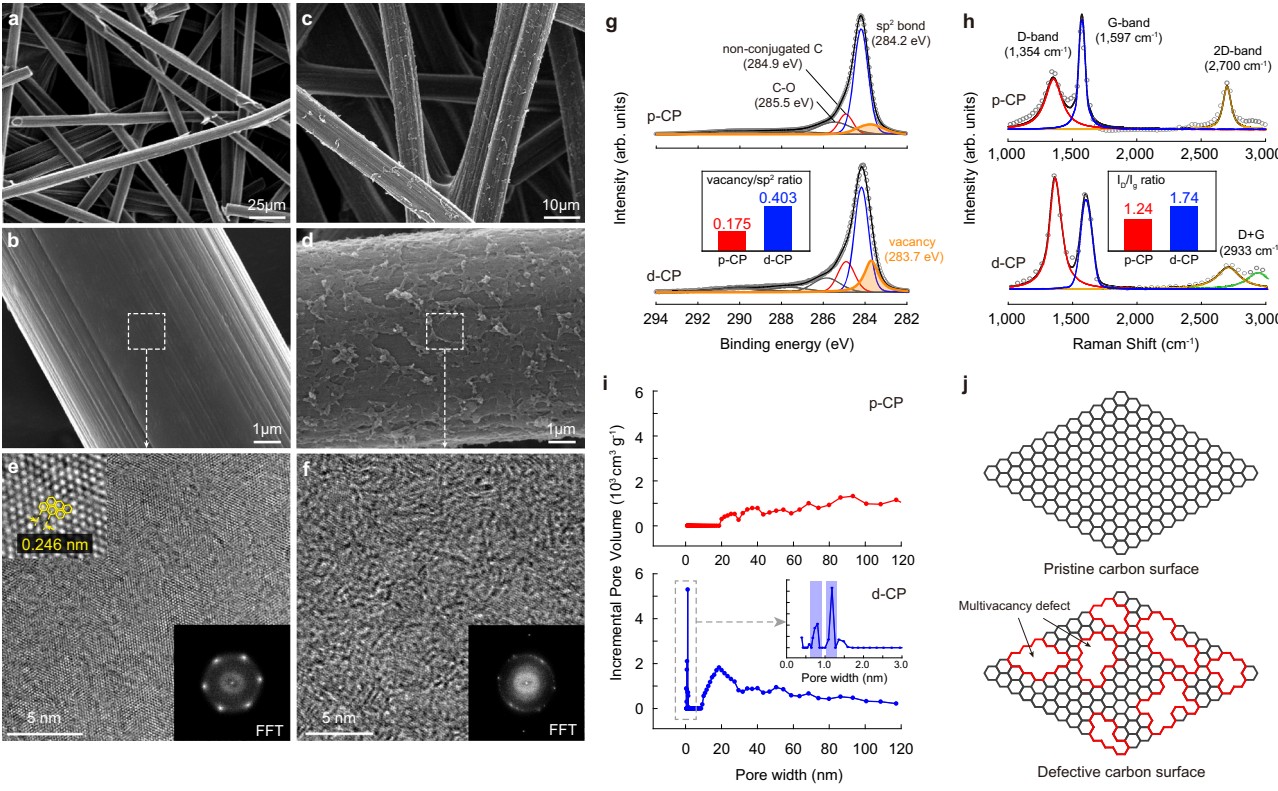

**Fig. 2 Atomistic structure of the defective carbon layer. a–d** SEM images of p-CP (**a**, **b**) and d-CP (**c**, **d**). **e**, **f** HR-TEM images and corresponding FFT patterns for p-CP (**e**) and d-CP (**f**). **g** C 1s HR-XRS spectra for p-CP and d-CP. The binding energy in XPS was calibrated with the C 1s $sp^2$ peak at 284.2 eV. (Inset: Comparison of the vacancy/$sp^2$ ratio determined by dividing the area of the vacancy peak by that of the $sp^2$ peak between p-CP and d-CP). **h** Raman spectra for p-CP and d-CP. (Inset: Comparison of $I_D/I_G$ ratio calculated by dividing the area of the D-band by that of the G-band between p-CP and d-CP) **i** BJH pore size distribution analysis for p-CP and d-CP. Inset is the magnified micropore region for d-CP. **j** Schematic illustrations of the atomistic structure of the p-CP and d-CP surfaces. The red line indicates the MV defects of d-CP.

region is a general characteristic of micropores (<2 nm). In the BJH pore size distribution analysis, d-CP exhibited an intense signal at 1.2 nm and a weak signal at 0.9 nm, which corresponded to 8 and 6 missing carbons, respectively. Based on the spectroscopic and pore size analysis, d-CP can be characterized as an MV defect-enriched surface, as shown in Fig. 2j.

**Defect-free SEI at the current collector–electrolyte interface.** The decomposition of the electrolyte on the anode current collector accompanies the evolution of the SEI. Because the structure of the SEI on the current collector plays a critical role in guiding subsequent Li electrodeposition in the anode-free configuration, the microstructure of the initial SEI on the current collector

should be controlled. In addition, because the consumption of electrolyte to form the SEI increases linearly with the surface area of the electrode, current collectors with expanded surface area, such as in the case of a 3D current collector, are hardly employed for anode-free batteries despite the advantage they provide, namely, lowering the effective current density for uniform Li deposition. Therefore, we sought to design a current collector/electrolyte interface that generates a uniform SEI microstructure and suppresses electrolyte decomposition.

Figure 3a shows the first cycle voltage profile of p-CP and d-CP for a galvanostatic reduction down to 0.0 V and subsequent oxidation up to 2.0 V. As an electrolyte, 1 M LiPF$_6$ ethylene carbonate (EC)/diethyl carbonate (DEC) (1/1 in volume) with 10 wt% fluoroethylene carbonate (FEC) and 1 wt% vinylene carbonate (VC) was used. The similar lithiation capacities of p-CP and d-CP indicated that the introduced defect layer had little effect on the reversible graphite areal capacity. In the potential window of 0–2.0 V, Li metal deposition did not occur; thus, the irreversible capacity originated from electrolyte decomposition and consequent SEI generation on each current collector. The initial Coulombic efficiency (CE) for d-CP was 87.4%, which was higher than that of p-CP (80.8%), despite its much larger electroactive surface area (Supplementary Fig. 2). According to linear sweep voltammetry (LSV) (Fig. 3b), d-CP also exhibited a much lower reduction current per electroactive surface area from electrolyte decomposition, demonstrating that the surface of d-CP was less active for the reduction of electrolyte. As a graphite electrode, d-CP showed higher CEs than p-CP for the Li intercalation/deintercalation process, verifying the decrease in electrolyte decomposition (Supplementary Fig. 3). The result is antithetical to previous perceptions that the irreversible capacity from SEI formation is nearly proportional to the specific area of the electrode material and that a defective atomic structure of carbon promotes considerable SEI formation[12,14,23].

The structure of the SEI formed on the p-CP and d-CP current collectors according to the charge voltage cutoff was analyzed through XPS. The composition of the inner SEI was observed after charging up to 1.0 V and the outer SEI after charging up to 0.0 V. As shown in Fig. 3c, the SEI composition of p-CP exhibited strong potential dependency: when lowering the cutoff voltage, the intensities of LiF and Li$_2$CO$_3$ decreased, and the intensities of C–O, C=O (components for lithium alkyl carbonate, R-OCO$_2$Li) and Li$_2$O increased, indicating a disparity in the SEI composition between the inorganic-rich inner and organic-rich outer SEI layers, which is consistent with a typical multilayer SEI model[14]. In sharp contrast, regarding d-CP (Fig. 3d), the XPS spectra for two different cutoff voltages were nearly identical, exhibiting Li$_2$CO$_{3-}$ and LiF-rich structures. This indicates that the early formed SEI on d-CP effectively suppressed further electrolyte decomposition. The F 1s spectra at 1.0 V demonstrates the formation of LiF for both p-CP and d-CP. However, for the cutoff voltage of 0.0 V, strong signals of Li$_x$PO$_y$F$_z$ and LiP$_x$F$_y$, indicative of the decomposition of PF$_6^-$, newly appeared for p-CP in contrast to the weak Li$_x$PO$_y$F$_z$ and LiP$_x$F$_y$ signals observed for d-CP. The XPS analysis suggested the chemical structures of SEI on p-CP and d-CP, as illustrated in Fig. 3e.

To investigate the thickness and microstructure of the SEI, TEM analysis was performed (Fig. 3f, g). The thickness of the SEI on d-CP (17.3 nm) was approximately one-third of that on p-CP (~50 nm), verifying that SEI formation by electrolyte decomposition was suppressed on d-CP. A closer look at the SEIs revealed that the SEI on p-CP had an irregular mosaic SEI structure and randomly dispersed organic and inorganic grains (indicated by orange circles), whereas d-CP had a uniform SEI structure. Considering that the mosaic structure originated from successive decomposition of different electrolyte components at different

potentials, the thin and uniform SEI on d-CP suggested self-limiting SEI formation at the early stage of electrolyte decomposition. In particular, the SEI on d-CP showed a homogeneous structure without any observable crystalline domains larger than a few nanometers. Based on the high self-limiting property of SEI formation on d-CP, we presumed that SEI components were formed sporadically without the growth of organic or inorganic grains, resulting in a uniform SEI structure rather than a grain-coalesced mosaic structure. The high uniformity of the SEI on d-CP can facilitate uniform Li$^+$ flux and diminish focused Li nucleation, which are requirements for uniform Li nucleation and growth (Fig. 3h, i).

**Alleviated electron transfer on MV defects**. To gain insight into the mechanism of thin and uniform SEI formation on d-CP, we investigated the change in the atomic and electronic structures of p-CP and d-CP during the initial lithiation before lithium metal deposition. Figure 4a compares the potential profiles in the voltage range from the open-circuit voltage (OCV) to 0.8 V for p-CP and d-CP. A larger capacity in the potential window was observed for d-CP than p-CP. The distinguished potential plateau at 1.2 V observed for p-CP corresponded to the decomposition potential of FEC. The atomic structural changes of p-CP and d-CP during the early stage lithiation were investigated via operando X-ray diffraction (XRD) measurements (Fig. 4b, c). The XRD image plots in the 24–30° region shed light on the lithiation process of the graphitic structure of carbon fiber. In contrast to the strong peak at 2θ of 26.5° from G(002) of p-CP, the weak intensities at ~26.5° for d-CP indicated the existence of a defective carbon layer. Regarding d-CP, the lithiated structures of LiC$_6$ and LiC$_{12}$ started to appear at lower lithiation capacities than those of p-CP, suggesting that relatively fast Li transport through the defective layer facilitated the lithiation of the inner graphitic carbon fiber. The structural evolution of the defect structure can be traced from the higher 2θ region. From the beginning of lithiation, a peak at 2θ = 43° appeared for d-CP and gradually grew with decreasing potential, indicating Li insertion into the defective structure of the graphite surface[24,25]. In contrast, the XRD pattern of p-CP was invariant in the potential range where solid-solution type lithiation at the 1L stage of initial graphite lithiation (Li$_x$C$_6$, x < 0.16)[26] occurred; the solid-solution type lithiation did not change the lattice structure of the graphite surface.

To trace the electronic structure along with the variation of the surface atomic structure, the work function, which is the difference between the vacuum level ($E_{vac}$) and Fermi level[27], was measured for p-CP and d-CP as the potential descended by using UV photoelectron spectroscopy (UPS, Supplementary Fig. 4). The work function of the intrinsic electrode surface of p-CP was 4.52 eV, which is coincident with the work function of graphite reported in previous studies[28], and that of d-CP was 4.80 eV (Fig. 4d). The Fermi levels of p-CP and d-CP increased with decreasing potential upon lithiation. Notably, the Fermi levels of d-CP were much lower than those of p-CP regardless of the degree of lithiation, indicating that the surface of d-CP had a lower electron energy level upon charging.

In line with the UPS result, DFT calculations also demonstrated that the MV-defective structure had a lower Fermi level than the non-defective surface during lithiation. Based on the 2D/G ratio from the Raman spectroscopy results and the pore size distribution, we constructed bilayer graphene (BLG) and MV defects in BLG (Supplementary Figs. 5 and 6). According to the calculations of the sequential lithiation of MV defects, a Li atom spontaneously penetrated through the defect hole, unlike other single or double vacancy defects, accepting the electrons of Li by the electron-deficient MV defect (Supplementary Figs. 7 and 8) and places

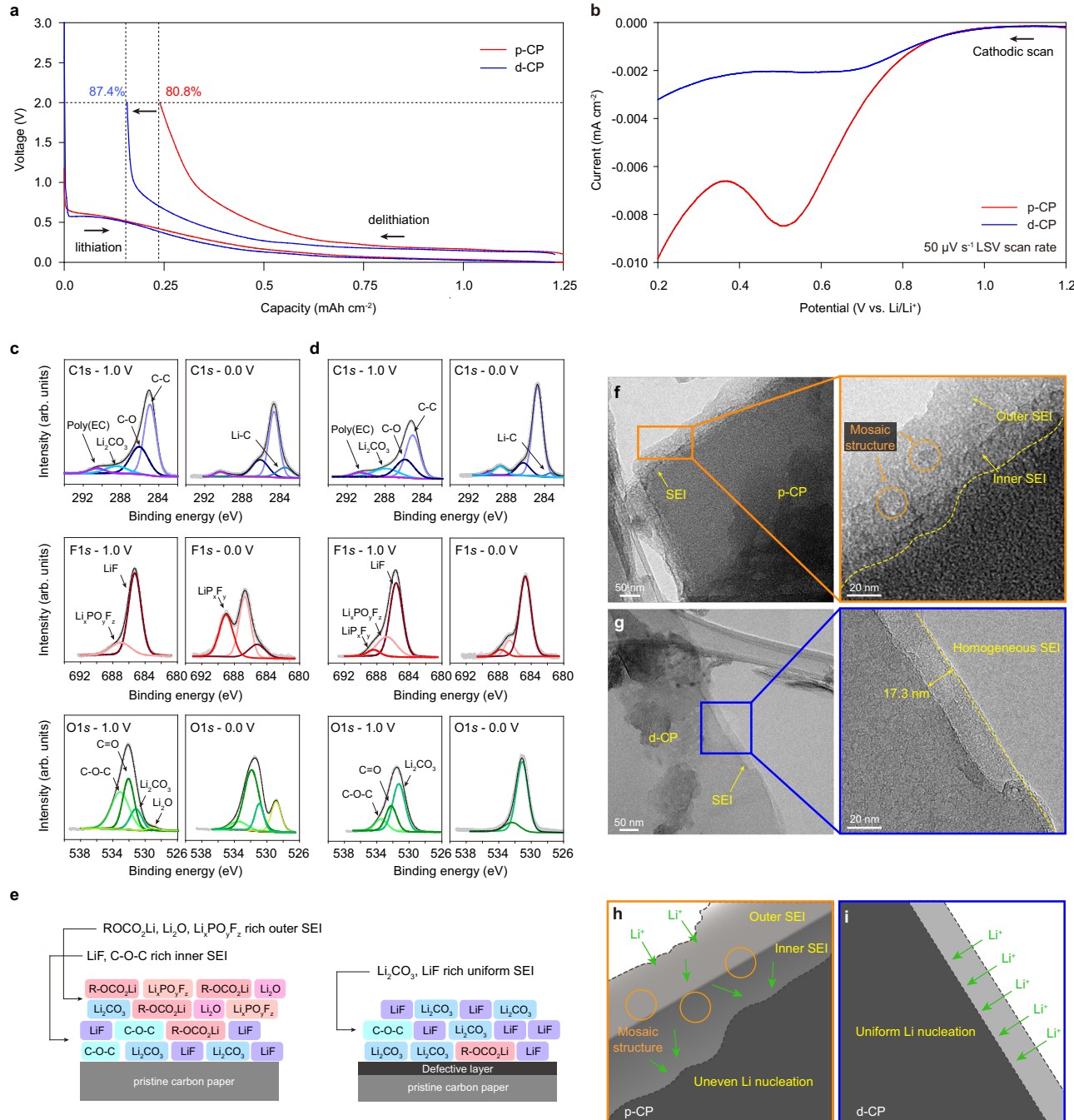

**Fig. 3 SEI formation on the multivacancy carbon current collector. a** Galvanostatic lithiation–delithiation curves for Li‖p-CP and Li‖d-CP cells in the range of 0.0 to 2.0 V at 0.1 mA cm$^{-2}$ for the quantification of irreversible capacity at the first cycle due to electrolyte decomposition at the current collector. **b** Current density from electrolyte decomposition based on the active surface area of the current collector (Supplementary Fig. 2a, b) during the LSV measurement (cathodic scan) at a 50 µV s$^{-1}$ scan rate for the Li‖p-CP and Li‖d-CP half-cells. **c, d** Ex situ XPS analysis of the SEI layer on p-CP (**c**) and d-CP (**d**). The remaining spectra were taken after galvanostatic reduction at 0.1 mA cm$^2$ down to 1.0 V (inner SEI), and the right spectra to 0.0 V (outer SEI). The binding energy was calibrated with the C–C bond peak of 285 eV in the C 1s spectra. **e** Suggested SEI composition on p-CP (left) and d-CP (right). **f, g** TEM images of the SEI layers for p-CP (**f**) and d-CP (**g**) after galvanostatic reduction to 0.0 V. **h, i** Schematic representation of the SEI microstructure formed on p-CP (**h**) and d-CP (**i**).

between the graphene layers with a gradual decrease in spin momentum (Supplementary Figs. 9 and 10) until a stoichiometry of Li$_3$C$_8$ (fully lithiated state) was reached. The calculated potential profile and Fermi levels of lithiated MV defect structures (Li-MVs) at different degrees of lithiation are shown in Fig. 4e. The potential change with lithiation accounted for the larger lithiation capacity of d-CP above 1.5 V vs. Li/Li$^+$. In accordance with the UPS results,

the calculated Fermi levels of the defect structure were much lower than those of BLG. These results indicated that the Fermi level of d-CP was lower than that of p-CP even though the defects were lithiated, and the electron-deficient properties of d-CP caused a change in the SEI structure on the surface.

Based on tunneling theory in quantum mechanics, the disparity in the SEI thickness between p-CP and d-CP can be

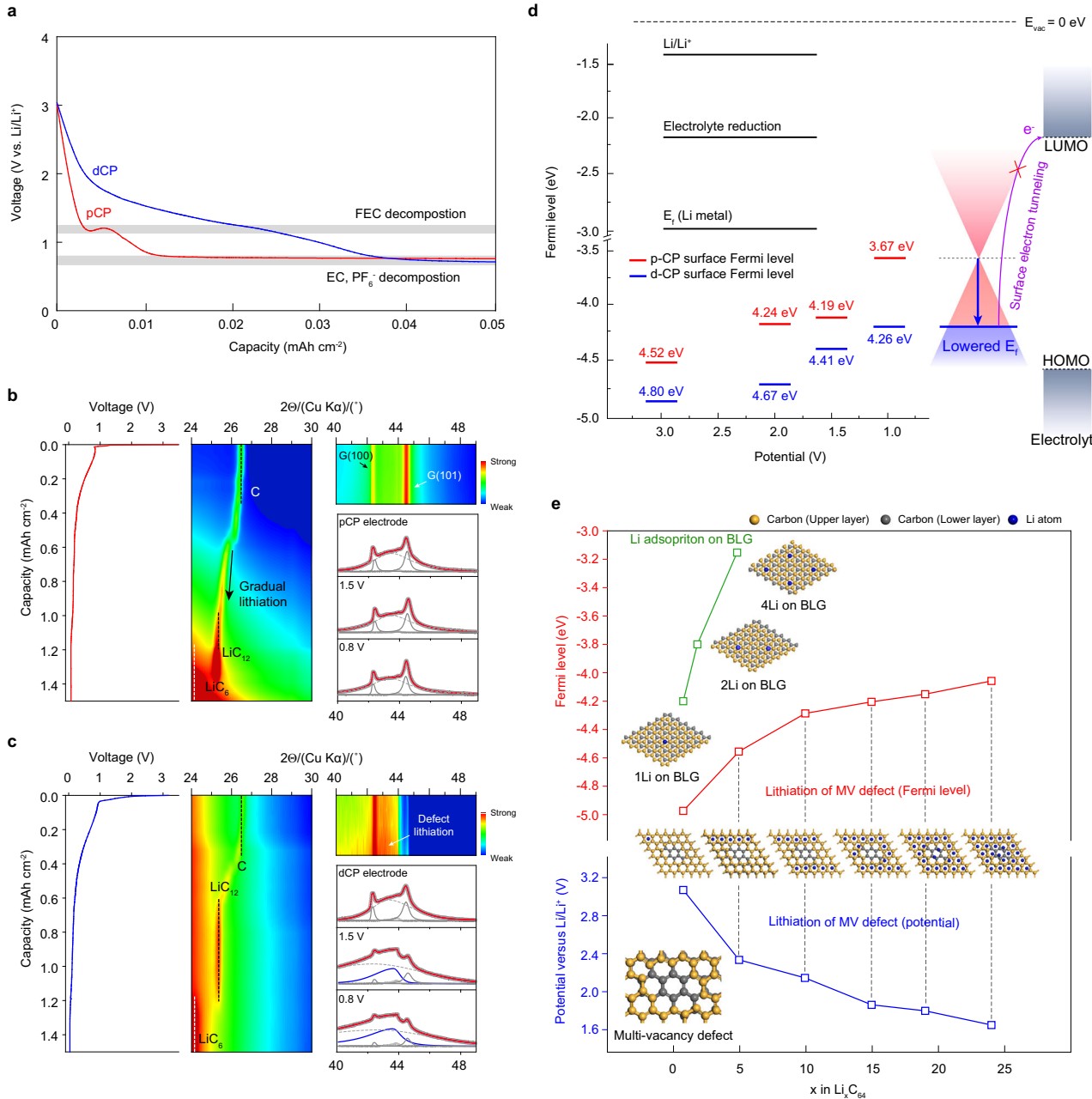

**Fig. 4 Atomistic and electronic structure of multivacancy defects during lithiation. a** Voltage profiles for the early stage of galvanostatic lithiation for p-CP and d-CP at 0.1 mA cm$^{-2}$ in the half-cell configuration. **b**, **c** Image plots of operando-XRD spectra along with the potential profile during the first lithiation and deconvoluted XRD patterns at the OCV, 1.5 V and 0.8 V for p-CP (**b**) and d-CP (**c**). The peaks from the lithiated defect structure (Li$_3$C$_8$) at $2\theta$ of 43° are indicated by blue lines. **d** Plots of the Fermi levels of p-CP and d-CP during lithiation from the OCV to 1.0 V (V vs. Li/Li$^+$), as determined from extrapolation of the secondary cutoff region in UPS. All the Fermi levels of Li/Li$^+$, redox associated with electrolyte reduction, Li metal, p-CP, and d-CP were calibrated at a vacuum state of $E_{vac} = 0$ eV. **e** Relaxed structures of lithiated BLG, MV, and fully lithiated MV (Li$_3$C$_8$), corresponding DFT calculations of the Fermi level, and redox potential of MV at various lithiation degrees. The Li, carbon in the upper layer, and carbon in the lower layer are shown in blue, yellow, and gray colors, respectively.

understood. The electron insulating property of the SEI depends on the SEI layer thickness ($d$) and Fermi level ($E_f$), as shown in Eq. (1)[23]:

$$T = \frac{16 E_f \Delta E_t}{\left(E_f + \Delta E_t\right)^2} e^{-\frac{4\pi d}{h}\sqrt{2m\Delta E_t}} \quad (1)$$

where $T$ is the electron tunneling probability for complete electron insulation (namely, $T = e^{-40}$),[14,23] $E_f$ is the Fermi level

of the electrode surface ($E_{vac} - \Phi_{electrode}$, where $\Phi_{electrode}$ is the work function), $\Delta E_t$ is the electron tunneling barrier (energy difference between the conduction band minimum of the SEI layer ($E_{CBM,SEI}$) and $E_f$), $m$ is the mass of an electron, and $h$ is Planck's constant. Equation (1) can be simplified as $T \sim e^{-d\sqrt{\Delta E_t}}$ by the Wentzel–Kramers–Brillouin approximation in quantum tunneling theory[23]. Since $\Delta E_t$ increases with decreasing $E_f$, d-CP with a lower $E_f$ can reach electroinsulation at a smaller $d$.

Therefore, the electron-deficient defective surface can alleviate electrolyte decomposition and limit SEI growth at initial lithiation.

**Uniform Li deposition triggered by MV defects.** Uniform Li deposition is a prerequisite for high-performance anode-free batteries. Fully lithiated MV defects can act as efficient nucleation sites due to their lithiophilic nature. We investigated the morphological evolution of Li nuclei formed on p-CP and d-CP at 1.0 mA cm$^{-2}$. Regarding p-CP (Fig. 5a), Li nuclei were sparsely populated (1.0 μAh cm$^{-1}$), and they grew normal to the surface (2.0 μAh cm$^{-1}$). The distribution of Li nuclei on p-CP would be determined by irregular Li$^+$ flux in the defect-rich SEI or grain boundaries of the basal plane with higher surface energy[29]. With further Li deposition (5.0 μAh cm$^{-1}$), Li nuclei clustered, suggesting the surface diffusion of Li adatoms or nuclei on the low-energy graphene surface and their coalescence by van der Waals interactions[30]. A focused investigation of a Li cluster (Fig. 5b) revealed that the agglomerated Li nuclei had poor regularity in the lattice structure, as observed in the TEM image, scanning transmission electron microscopy (STEM), and the pattern of rings in the FFT images. The aggregative growth behavior corresponded to the Volmer–Weber growth mode, which depicted a situation where metal-to-metal interactions were stronger than metal-to-substrate interactions[30,31].

In sharp contrast, regarding d-CP, Li nuclei were densely populated and homogeneously distributed on the surface (Fig. 5c), suggesting that the defect-free SEI provided uniform Li$^+$ flux to the electrode surface[13,32] and that the lithiated MV defects acted as favorable nucleation sites. Further Li deposition led to the lateral growth of nuclei and the merging of nearby nuclei to form a planar Li deposit (5.0 μAh cm$^{-1}$). This observation implied that metal-to-substrate interactions were stronger than metal-to-metal interactions, which corresponded to the Frank-van der Merwe growth mode[31]. According to the TEM, STEM, and FFT analyses (Fig. 5d), the Li nuclei formed on d-CP had a monocrystalline surface facet, which confirmed the prevention of surface diffusion and aggregation of Li nuclei by the MV defect structure.

The electrochemical analysis further confirmed the distinguished Li nucleation/growth behaviors between p-CP and d-CP. As shown in Supplementary Fig. 11, the Li nucleation behaviors can be successfully analyzed by the Scharifker and Hills theory. Regarding p-CP and d-CP, the Li deposition process showed an instantaneous nucleation mode. The nuclei number densities calculated from the CA data were almost two orders of magnitude higher for d-CP than for p-CP (Supplementary Fig. 11c), which is consistent with the SEM observations.

To obtain an understanding of the uniform Li deposition on d-CP, we performed a DFT analysis of electronic properties, including the partial density of states (pDOS) (Fig. 5e and Supplementary Fig. 12), partial charge distribution (Supplementary Fig. 13), and adsorption energies of the Li adatom. In Fig. 5e, the relaxed structures and pDOS spectra of single Li adsorbed on non-defective BLG, lithiated MV defects, and lithiated MV defects with four preadsorbed Li atoms are provided. The DOS spectra of Li atoms on perfect BLG featured sharp pDOS of the Li $s$ and $p$ orbitals. The DOS of the C $s$ and $p$ orbitals on perfect BLG were invariant with Li adsorption (Supplementary Fig. 6a and Supplementary Fig. 12a). These results indicated the absence of electron transfer between BLG and Li. In contrast, regarding Li atoms on lithiated MV defects, broadening of the DOS of Li and C $s$ and $p$ was observed, indicating the hybridization of Li orbitals with C orbitals by electron transfer from the electropositive Li atoms to electron-deficient MV defects (Supplementary Fig. 12b–d). In the same vein, the partial charge distribution of

Li adatoms and carbon atoms of the edge of the MV defect also supported electron transfer between them (Supplementary Fig. 13). Therefore, the facile Li nucleation on d-CP was attributed to the strong orbital hybridization between Li and MV defects. The arrangement of clusters of Li adatoms on BLG and MV defects shown in Fig. 5f, g supported the propensity of lateral Li growth for d-CP and vertical Li growth for p-CP. The five successively adsorbed Li atoms on d-CP were distributed laterally on the lithiated MV defect, whereas they exhibited vertical aggregation on the BLG surface.

The comparison of the adsorption free energy[33] of Li atoms on various atomic structures explained the strong Li-to-substrate interaction of d-CP (Fig. 5h). The adsorption free energy was highest on Li(111) (1.89 eV) among various Li facets, which is identical to previous studies[34,35]. The adsorption free energy for BLG was 1.73 eV, which is a lower value than that for Li(111), supporting the Volmer–Weber growth mode for p-CP. However, the adsorption free energy for the center of the lithiated MV defect was 3.13 eV, which is larger than those for any Li facets, which is in good agreement with the Frank-van der Merwe growth mode observed for d-CP. Additionally, the adsorption energy at the center of the MV defect was much larger than that when the Li atoms were located surrounding the MV defect (Supplementary Fig. 14), explaining the hindered surface diffusion of Li nuclei for d-CP (Supplementary Fig. 15).

**Cycling stability of the MV-defective carbon current collector.** We investigated the reversibility of Li plating/stripping for Cu, p-CP, and d-CP with a conventional carbonate electrolyte (1 M LiPF$_6$ EC/DEC+10% FEC+1% VC). Figure 6a compares the CEs of Li|| Cu, Li||p-CP, and Li||d-CP half-cells during cycling at 2.0 mA cm$^{-2}$ and 5.0 mAh cm$^{-2}$. The voltage profiles during cycling are shown in Supplementary Fig. 16a–c. 2D Cu and 3D p-CP were compared to assess the influence of the reduced effective current density due to the use of the 3D structure. The Li||Cu cell showed an abrupt drop in CE at ~20 cycles with an average CE of 93%. The cell failure was attributed to the localized formation of a thick, porous layer (Supplementary Fig. 16d, e). The Li||p-CP cell operated for 60 cycles with a 92.1% average CE, indicating a better cycling performance than the 2D Cu current collector. However, from the ex-situ postmortem electrode measurements and analysis, the formation of a dead (electronically disconnected) Li layer on p-CP was observed to be similar to Cu (Supplementary Fig. 16f, g). The Li||d-CP cell exhibited high cycling stability for over 170 cycles with a 98.6% CE. There have been few reports of a high CE value and cycling stability at a highly curtailed capacity of 5.0 mAh cm$^{-2}$ in carbonate electrolyte-based half-cells (Supplementary Table 2). Contrary to the Li||Cu and Li||p-CP cells, the cell failure of Li||d-CP was not caused by the d-CP electrode but by the depletion of the Li-metal counter electrode; the disassembled Li||d-CP cell after cell failure showed the depletion of Li at the counter electrode and the preservation of the porous structure in d-CP without a thick dead (electronically disconnected) Li layer (Supplementary Fig. 16h, i). A rate capability test for p-CP and d-CP (Supplementary Fig. 16j) showed that d-CP can stably operate at 14 mA cm$^{-2}$ in contrast with the rapid capacity fade when cycling p-CP above 10 mA cm$^{-2}$.

As another measure of the reversibility of the Li electrode, a Li|| Li symmetric cell was constructed with Li-electroplated current collectors; Li was electrodeposited on Cu, p-CP, and d-CP at 0.2 mA cm$^{-2}$ and at 8 mAh cm$^{-2}$, respectively. The symmetric cells were cycled at 2 mA cm$^{-2}$ and at a Li utilization of 25% (2 mAh cm$^{-2}$). As provided in Supplementary Fig. 17, the d-CP symmetric cell maintained its smallest overpotential for more than 400 cycles, while the Cu and p-CP cells showed irregular voltage spikes at 45 cycles and 42 cycles, respectively. Consistent

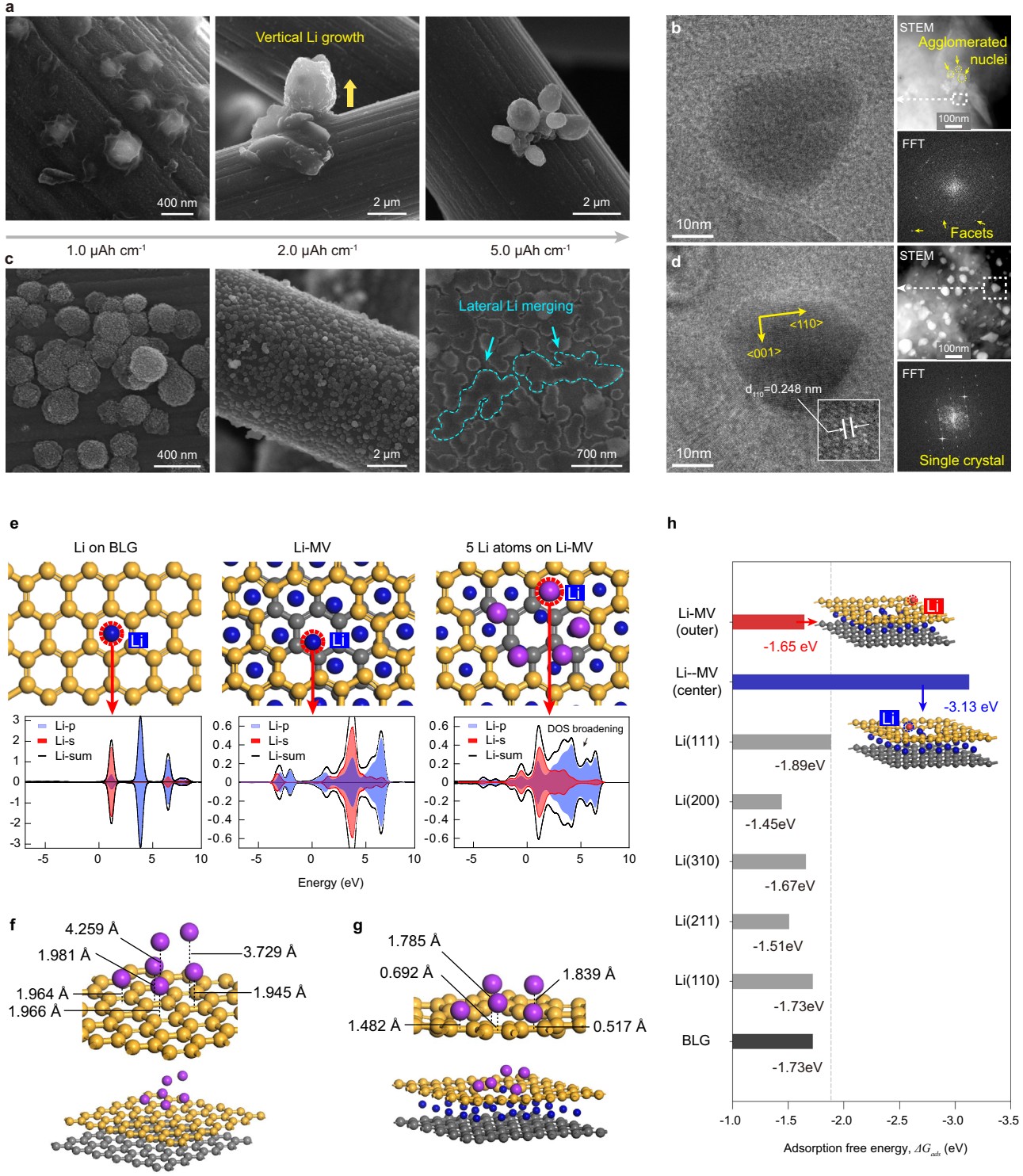

**Fig. 5 Mechanistic analysis of the morphological propensity of Li nucleation and growth on MV defects. a** SEM images of the Li deposit morphology on p-CP after Li deposition at 1.0 mA cm$^{-2}$ for 3.6, 7.2, and 18 s, which correspond to 1, 2, and 5 µAh cm$^{-2}$, respectively. **b** TEM, STEM, and FFT images of Li nuclei on p-CP at 1.0 mA cm$^{-2}$ and 1 µAh cm$^{-2}$. **c** SEM images of the Li deposit morphology on d-CP after Li deposition at 1.0 mA cm$^{-2}$ for 3.6, 7.2, and 18 s, which correspond to 1, 2, and 5 µAh cm$^{-2}$, respectively. **d** TEM, STEM, and FFT images of Li nuclei on d-CP at 1.0 mA cm$^{-2}$ and 1 µAh cm$^{-2}$. **e** Relaxed structures and pDOS results of a single Li atom adsorbed on perfect BLG, Li-MV, and Li-MV with four preadsorbed Li atoms. **f**, **g** Relaxed configurations of Li atom clusters absorbed on perfect BLG and Li-MV (Li$_3$C$_8$). The lithiated Li and Li clusters, carbon in the upper layer, and carbon in the lower layer are shown in blue, purple, yellow, and gray colors, respectively. **h** Adsorption free energies of Li atoms on the outer and center of Li-MV, various facets ((111), (200), (310), (211), (110)) of Li metal, and perfect BLG.

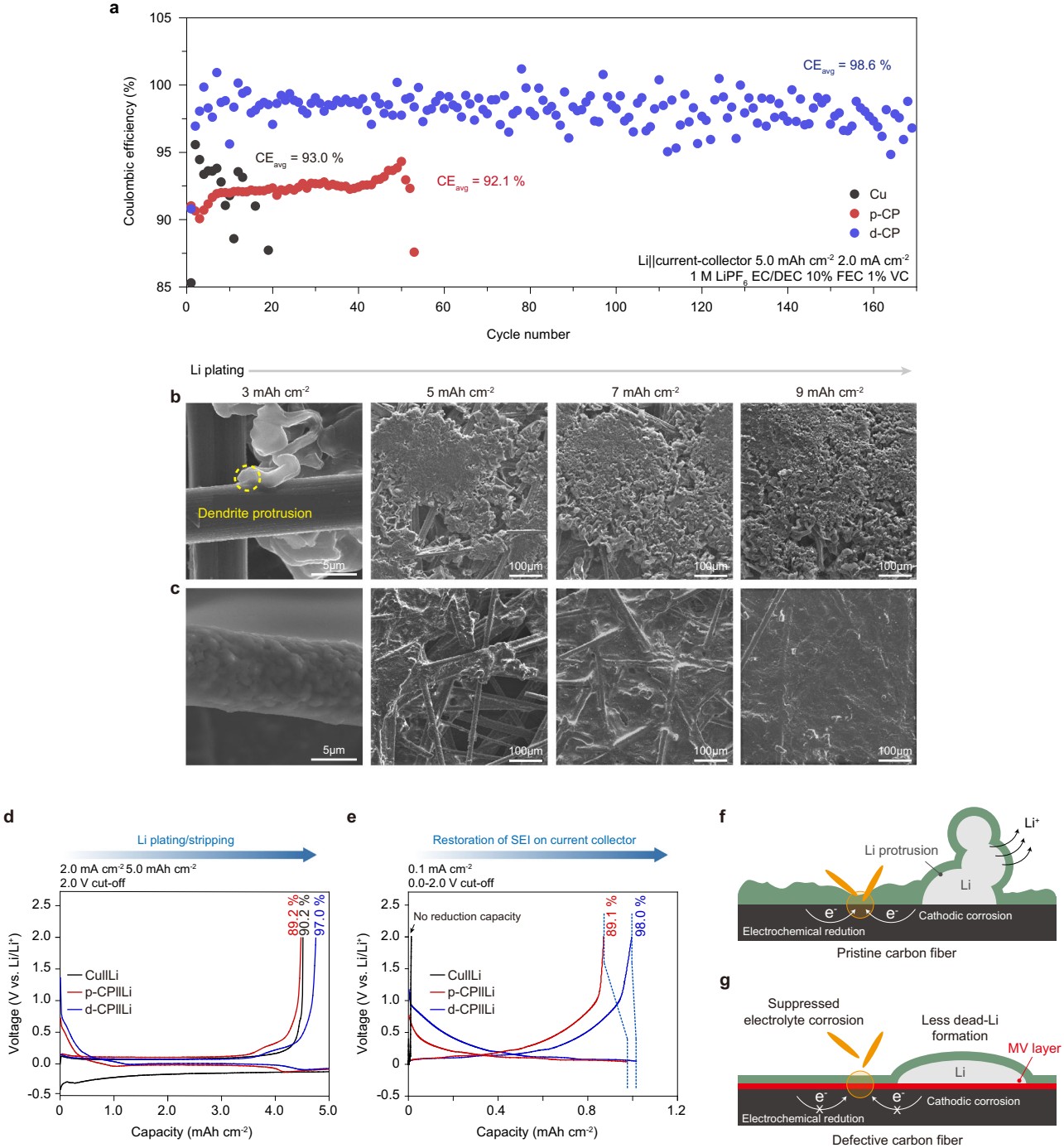

**Fig. 6 Reversibility of Li plating/stripping on the defective carbon surface. a** Plots of CE as a function of cycle number for Cu, p-CP, and d-CP current collectors in a half-cell configuration at a current density of 2.0 mA cm$^{-2}$ and a capacity of 5.0 mAh cm$^{-2}$. **b, c** Li deposition morphologies for p-CP (**b**) and d-CP (**c**) at various plating capacities from 3.0 to 9.0 mAh cm$^{-2}$. **d, e** Quantification of the irreversible capacity caused by Li metal plating/stripping and sequential restoration of damaged SEI on the current collector surface. SEI restoration was performed by galvanostatic cycling in the range of 0.0 and 2.0 V at a current density of 0.1 mA cm$^{-2}$. **f, g** Schematics of the irreversible loss of Li and the electrolyte on the p-CP surface (**f**) and the proposed mechanism for preventing irreversibility on the d-CP surface (**g**).

with the uniform nucleation of Li on MV defects, the d-CP cell showed the lowest deposition overpotential of 15 mV, which was lower than the Cu cell value of 55 mV and p-CP cell value of 28 mV.

The evolution of the Li morphology during Li electrodeposition at 1.0 mA cm$^{-2}$ with the carbonate electrolyte was observed for p-CP and d-CP. Figure 6b shows the SEM images of p-CP at various Li deposition capacities (3, 5, 7, 9 mAh cm$^{-2}$). The

localized protrusion of Li at the surface of p-CP found at 3 mAh cm$^{-2}$ triggered suffusion of the Li deposit on the top surface of p-CP at larger deposition capacities. This result indicated that the top Li deposition prevented the utilization of the structural tridimensional porous network of p-CP (Supplementary Fig. 16k). In sharp contrast, regarding d-CP (Fig. 6c), its carbon fibers were uniformly covered by plated Li at 3 mAh cm$^{-2}$ due to uniform Li nucleation and the lateral growth driven by the MV defect.

Further deposition led to the connection of the Li deposits to those on adjacent fibers. At 9 mAh cm$^{-2}$, a dendrite-free and smooth Li morphology was observed without the top plating of Li (Supplementary Fig. 16l). The absence of the top plating for d-CP suggested that the reduced nucleation overpotential (Supplementary Fig. 16a–c) by MV defects can induce homogenous initial nucleation in the interior of d-CP. Unlike p-CP, top plating was not observed for d-CP even at the Li deposition capacity (16 mAh cm$^{-2}$), corresponding to the full accommodation of its pores (Supplementary Table 3 and Supplementary Fig. 18).

The origin of the irreversible capacity in anode-free batteries was quantitatively analyzed by investigating a single Li plating/stripping cycle and the subsequent partial cycling of carbon lithiation/delithiation (Fig. 6d, e). The irreversible capacity of the first cycle originated from the formation of an SEI on the current collector and the deposition of Li and dead (electronically disconnected) Li, whereas that of the second partial cycle should reflect the restoration of the SEI on the current collector, which was damaged during the first cycle. The capacity loss from Li plating/stripping was 0.490 mAh cm$^{-2}$ (CE of 90.2%) for Cu and 0.541 mAh cm$^{-2}$ (CE of 89.2%) for p-CP, which indicated that a larger amount of Li was lost with p-CP and that SEI formation on its large surface area offsets the positive effect of a reduced current density. The irreversible capacity at the second cycle was 0.143 mAh cm$^{-2}$ for p-CP, which meant that the capacity loss by the restoration of the fractured SEI was significant. In contrast, regarding d-CP, the initial irreversible capacity during Li plating/stripping was 0.152 mAh cm$^{-2}$ (CE of 97.0%), which is approximately one-fourth of that of Cu or p-CP. Remarkably, the second irreversible cycling capacity due to the restoration of the SEI was 0.02 mAh cm$^{-2}$; this result indicated that the SEI on d-CP may not be significantly damaged due to the uniform Li plating/stripping and, even if damaged, the fractured SEI can be repaired with a minimal amount of electrolyte due to the low Fermi level of d-CP. Overall, the interfacial instability of p-CP can be attributed to the (1) thick and inhomogeneous SEI formation on p-CP, (2) whisker-shaped Li morphology that is prone to Li corrosion by an electrolyte, and (3) fracture of the SEI triggering the cathodic corrosion of Li, as described in Fig. 6f. In contrast, regarding d-CP, lateral Li growth reduced the generation of inactive Li, and its insulating surface prevented cathodic Li corrosion, thereby increasing the reversibility of galvanostatic cycling (Fig. 6g).

**Electrochemical performance of an anode-free Li-metal cell**. To evaluate the practical applicability of the MV-defective carbon current collector, we assembled anode-free coin cells by using a high-nickel NCM811 cathode with a high areal capacity (>4.0 mAh cm$^{-2}$), employing a flooded electrolyte design (electrolyte to capacity ratio (E/C) of 23.8 μl mAh$^{-1}$) or a lean electrolyte design (E/C of 4.0 μl mAh$^{-1}$), and various current collectors of Cu, p-CP, and d-CP. In our anode-free cell, since the lithiation/delithiation of the graphitic carbon current collector and plating/stripping of lithium occur, carbonate solvents (EC, DEC, and FEC) were selected to prevent the exfoliation of graphite rather than ethereal solvents[36]. In addition, carbonate electrolytes enable the use of NCM cathodes due to their high oxidation stability. LiPF$_6$ was chosen due to its high ionic conductivity in carbonate solvents. VC (2%) was added to help the formation of a stable SEI and cathode-electrolyte interphase (CEI)[37]. The voltage profiles of the anode-free cell with p-CP or d-CP (Fig. 7a) showed a distinctive carbon lithiation step at the beginning of charging (shaded in blue), as observed for the half-cells. As displayed in Fig. 7b, the operation current density and the areal capacity of the

anode-free cells with d-CP were higher than the previously reported values for liquid electrolyte-based anode-free cells[2,38–47].

Figure 7c shows the cycling performance of the anode-free cells at practical charging and discharging current densities of 2 mA cm$^{-2}$ (0.48 C) under flooded and lean electrolyte conditions. The Li-plating capacity and intercalation capacity for the operation were 3.70 and 0.30 mAh cm$^{-2}$ for p-CP, respectively, and 3.68 and 0.32 mAh cm$^{-2}$ for d-CP, respectively (Supplementary Fig. 19). The anode-free cell with the Cu current collector under flooded conditions showed 80% capacity retention before 10 cycles. A slightly improved cycling stability of 80% capacity retention at 20 cycles with a mitigated capacity fade rate was obtained with p-CP, likely due to the decreased effective current density and confined volume change[48]. With the d-CP current collector, the cycling stability of the anode-free cell was improved to 90% and 56% capacity retention at 50 cycles and 100 cycles, respectively, demonstrating its efficacy in stabilizing the anode-free electrode in a full cell. Under lean electrolyte conditions, the Cu and p-CP cells showed a faster capacity fade and significant capacity fluctuation compared with the flooded condition because of electrolyte depletion in the early cycles. In particular, p-CP exhibited a larger decrease in cycling stability than the flooded system under lean electrolyte conditions because of the difficulty in preserving the electrolyte wetting of the porous 3D current collector due to the continuous consumption of electrolytes. In strong contrast, the d-CP cell under lean electrolyte conditions exhibited 90% retention at 50 cycles. The rapid decrease in discharge capacity after 60 cycles was attributed to electrolyte depletion (Supplementary Fig. 20).

The cell-level energy densities with different current collectors were estimated based on the 0.1 C charge/discharge voltage profiles. In this calculation, we considered the single cell consisting of one sheet of a double-side-coated NCM cathode, one sheet of a fully charged anode (current collector + deposited Li), two sheets of a separator, and an electrolyte (Supplementary Table 4 and Supplementary Fig. 21). The cell with d-CP can deliver specific energy of 349.1 Wh kg$^{-1}$ and an energy density of 1084.7 Wh l$^{-1}$. Assuming void-free Li deposition (5 μm per 1.0 mAh cm$^{-2}$ Li), the cell with the Cu current collector had lower specific energy (338.2 Wh kg$^{-1}$) but a higher energy density (1304 Wh l$^{-1}$) compared with the d-CP cell. However, when the increased electrode thickness, due to the growth of Li dendrites and the formation of a thick, porous layer, was reflected in the calculation, the energy density of the Cu cell after 10 cycles decreased to 790.8 Wh l$^{-1}$, which is contrasted by the invariant energy density of d-CP (Supplementary Fig. 22). From a practical point of view, the use of carbon paper possesses a risk of mechanical deterioration at a much larger Li-plating capacity or after significant bending, hindering its use for winding-type cells. However, the MV defect strategy could be used to modify the surfaces of various current collectors with higher mechanical strength and flexibility.

## Discussion
We demonstrated the dual functionality of an electron-deficient MV defect by experimental and computational analyses. Compared with a non-defective surface, the MV defect has a different electronic structure with a low Fermi level due to its electron deficiency. On the MV defect-enriched surface, a thin and defect-free SEI layer was formed, and electrolyte decomposition was alleviated. The uniform SEI structure and strong Li–C orbital hybridization promoted uniform Li nucleation and lateral growth, resulting in a densely packed Li morphology. The MV-defective carbon current collector achieved high reversibility of Li plating/stripping at 5.0 mAh cm$^{-2}$ and 2.0 mA cm$^{-2}$ even with a

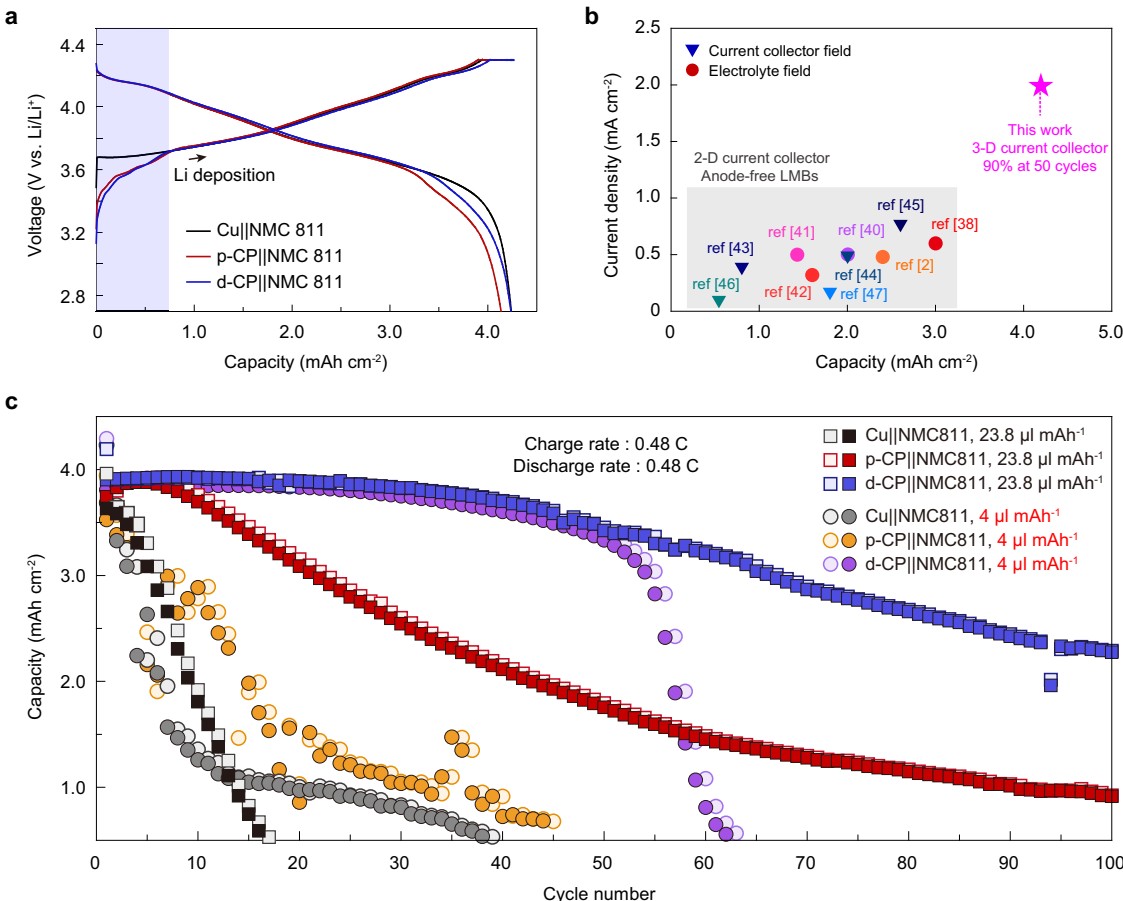

**Fig. 7 Electrochemical performance of anode-free coin cells with d-CP under practical conditions. a** Voltage profiles of the anode-free NCM811 full cell with Cu, p-CP, or d-CP during the charge at 0.1 C, sequential constant voltage charging at 4.3 V (cutoff current: 0.02 C), and discharge. **b** Comparison of the areal capacity and current density that ensures stable operation in this work and previous works. **c** Plots of charge and discharge capacity as a function of cycle number for anode-free NCM811 full cells with 1 M LiPF$_6$ EC/DEC+10% FEC+1% VC under flooded cell conditions (E/C ratio of 23.8 μl mAh$^{-1}$, square symbols) and lean electrolyte conditions (E/C ratio of 4.0 μl mAh$^{-1}$, circle symbols). The flooded and lean electrolyte cells were cycled at a charge/ discharge rate of 0.48 C (2 mA cm$^{-2}$) in a potential window of 2.7–4.3 V.

carbonate electrolyte. The introduction of MV defects significantly improved the cycling stability of an anode-free cell with high areal capacity (4.2 mAh cm$^{-2}$) operated for 50 cycles under lean electrolyte conditions (E/C of 4.0 μl mAh$^{-1}$) with 90% capacity retention. Based on the model system provided herein regarding how the electronic structure of the current collector controls the SEI microstructure and Li nucleation, a rational design of the surface electronic structure for advancing anode-free Li-metal batteries is presented.

## Methods

**Synthesis of the MV-defective carbon current collector**. An MV-defective carbon current collector (d-CP) was synthesized by the solution-synthesis method. The surface of carbon fiber paper (HCP010N, Shanghai Hesen Electric Co., Ltd., USA) was oxidized by oxygen plasma treatment. Zinc nitrate hexahydrate (5.95 g, 1 eq.) and 2-methyl imidazole (13.16 g, 8 eq.) were separately dissolved in 100 ml of methanol. The surface-oxidized carbon fiber paper was immersed in a mixture of Zn$^{2+}$ (15 ml) and 2-methyl imidazole (15 ml) solutions for 12 h to coat ZIF-8 on its surface. The samples were then thoroughly washed many times with ethanol to remove any remaining solvents. The ZIF-8 coating procedure was repeated two times. ZIF-8@p-CP was dried under vacuum overnight. Then, to form d-CP, it was carbonized with a temperature ramp rate of 5 °C min$^{-1}$ to 1000 °C in a N$_2$ atmosphere and held for 5 h. The carbonized d-CP was immersed in 2 M hydrochloric acid with stirring at room temperature for 6 h, followed by rinsing with distilled deionized water (DI) several times and drying under vacuum at 70 °C for 7 h.

**Electrochemical tests**. Electrochemical half-cell tests for the current collectors (16 mm in diameter) were conducted with a CR2032-type coin-cell using Li foil

(16 mm in diameter, 450 μm in thickness, Honjo Metal) as a counter electrode, a Celgard 2400 PP separator, and 1 M LiPF$_6$ EC/DEC (1/1 in volume, Panax) +10% FEC (Alfa Aesar) +1% VC electrolyte (100 μl). After a precycle at 0.2 mA cm$^{-2}$ between 2.0 and 0 V, galvanostatic cycling was conducted at 5 mA cm$^{-2}$ and 2 mA cm$^{-2}$ with a stripping potential cutoff of 1.0 V. A symmetric cell was assembled with two pieces of Li-plated copper (16 mm in diameter) or carbon current collectors. Li plating on the current collectors was carried out with 1 M LiTFSI DOL/DME (1/1 in volume, Panax) + 1% LiNO$_3$ (Sigma Aldrich) electrolyte at 0.1 mA cm$^{-2}$ with a 16 mm Li counter electrode. After the electroplating process, the Li-plated electrode was rinsed with DME (Sigma Aldrich, 99.5%, briefly washed with 200 μl DME in syringe). With the two prelithiated electrodes and 100 μl of 1 M LiTFSI DOL/DME v/v 1:1 1% LiNO$_3$ electrolyte, symmetric cells were assembled. The anode-free cell tests were carried out with a 2032 coin-type cell using a LiNi$_{0.8}$Mn$_{0.1}$Co$_{0.1}$O$_2$ (12 mm in diameter, NCM811:Super P:PVDF 96:2:2 mass ratio) cathode (20.79 mg cm$^{-2}$ of active materials, areal capacity of cathode is 4.2 mAh cm$^{-2}$ at 0.1 C, 0.1 C = 0.42 mA cm$^{-2}$), a current collector (12 mm in diameter, Cu, p-CP, d-CP), a Celgard 2400 PP separator, and 1 M LiPF$_6$ EC/DEC+10% FEC+1% VC electrolyte in a voltage range of 2.7–4.3 V at 0.48/0.48 C (charge/discharge). The NCM cathode was calendared by a built-in roll press in an Ar-filled glovebox before use, resulting in a coated layer (without current collector) thickness of 74 μm and density of 2.93 g cm$^{-3}$. After the constant current charging step, a constant voltage at 4.3 V was set for potentiostatic charging until the current density reached 0.02 C. Two precycles were conducted at 0.1/0.1 C (charge/ discharge). All the cell assembly processes were conducted in an Ar-filled glovebox (H$_2$O < 0.1 ppm, O$_2$ < 0.1 ppm). A WBCS3000L battery tester was used for the cell tests. All electrochemical tests were carried out in a cycling chamber and controlled at 25 °C.

**Structural characterization**. An SEM analysis was performed using field-emission scanning electron microscope (Sirion by FEI) with a 10 kV accelerating voltage and 5.0 mm working distance (Z-height). To observe the morphology of Li nucleation and growth, the cells were disassembled after Li plating (plated capacities are stated

in the results text) and loaded on an SEM holder in an Ar-filled glovebox. TEM and TEM-EDX analyses were carried out using a transmission electron microscope (Tecnai G2 F30) with a 200 kV accelerating voltage. The SEI formed on the current collectors (SEI was formed by the 1st cycle of the half-cell at a voltage cutoff of 0–2.0 V) was ground and dispersed in DEC (Sigma Aldrich, 99 %). The upper solution was sampled with a micropipette and loaded onto the TEM grid. The Raman spectra were measured using a Raman spectroscope (HORIBA Jobin Yvon, France) with a confocal microscope equipped with a solid-state crystal laser (wavelength of 514.5 nm and excitation energy of 2.41 eV) and a spot size of 0.5 mm. $N_2$ adsorption and desorption isotherms were obtained at 87 K on a Micromeritics 3Flex surface characterization analyzer. UPS analysis of the electrode work function with potential variation was conducted using in situ X-ray photoelectron spectroscopy (Axis-Supra by Kratos) with a He I X-ray source ($hv = 21.2$ eV). The samples for the UPS analysis were prepared by galvanostatic lithiation at 50 μA cm$^{-2}$ with various cutoff voltages (2.0 V, 1.5 V, and 1.0 V vs. Li/Li$^+$ in 1 M LiPF$_6$ EC/DEC), rinsed with dimethyl carbonate (DEC), and dried at 60 °C overnight in an Ar-filled glovebox before measurement. Since the UPS was very sensitive to the surface condition, to completely block the electrode from the atmosphere, the experiment was conducted on an instrument in which the UPS chamber and the glovebox were directly connected. To obtain the secondary cutoff region, a 9.0 V bias was applied to the samples. The Fermi level of the samples was aligned with the gold (Au) reference metal. For the work function calculation, the UPS data of the low kinetic energy cutoff region (secondary cutoff region) were extrapolated, and intercept points with the baseline were subtracted from the incident photon energy ($\Phi_{electrode} = 21.2$ eV – (binding energy of the intercept point)). The SEI composition and defective structure of the electrodes were analyzed by X-ray photoelectron spectroscopy (XPS, K-alpha by Thermo VG Scientific) with an Al X-ray source ($hv = 1486.7$ eV). The binding energies obtained from the XPS analysis were calibrated with the C–C bond peak at 284.8 eV in the C 1s spectra. An operando-XRD analysis was performed using an in situ battery cell X-ray diffractometer (R-Axis IV by RIGAKU) with a transmission Cu Kα X-ray source. The test cell had a 1.5 mm hole in the coin-cell cap and bottom parts, which were sealed by transparent mylar film and epoxy 4460.

## Data availability

The authors declare that all data supporting the findings of this study are included within the paper and its Supplementary Information. The source data for Figs. 4e and 7b are provided with the paper.

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

## Acknowledgements

This work was supported by LG Energy Solution, the KAIST Institute for Nano-Century (KINC), and by the Technology Development Program to Solve Climate Changes through the National Research Foundation of Korea (NRF), which is funded by the Ministry of Science, ICT (2018M1A2A2063807).

## Author contributions

H.K., J.-H.L., and H.-T.K. conceived the concept of the electron-deficient current collector and designed this work. These authors contributed equally: H.K. and J.-H.L. H.K. carried out the experimental planning, electrochemical measurements, characterization, and data analysis; J.-H.L. performed material synthesis, Raman spectroscopy, BET, and DFT calculations; Y.R., J.B., D.S., J.K.Y., H.J.H., and J.Y.K. contributed to the experimental design and discussion of the results; H.K. and H.-T.K. wrote the manuscript, and H.-T.K. supervised this work. All authors commented on the manuscript.

## Competing interests

The authors declare no competing interests.
