## [Peer Review File · Nature Communications]

REVIEWER COMMENTS

Reviewer #1 (Remarks to the Author):

The authors report on an interesting approach to enable anode-free lithium metal batteries. Anode-free lithium metal batteries is the ultimate as it has no excess lithium and provides highest energy density and low cost. However, liquid electrolytes generally react with the current collector and thus, modifying the current collector is a critical challenge being addressed through several approaches. Authors use a NW defect-enriched C-based current collector that provides good Li nucleation, growth of Li. The idea is novel, but authors should consider the comments below to improve their work:

- 1) The analysis shown in Fig. 5 is good, but the reference state as Li bulk would be better as that automatically provides a voltage scale. See for example, ACS Energy Lett., 2019, 4, 2952-2959.
- 2) What happens to the surface diffusion barrier for these systems? Do they observe any BEP relationship with the adsorption strength?
- 3) In Fig 7, the authors should compare against a few other data points from PNAS 2020 117 (44) 27195-27203 and the associated data points there.
- 4) Can the authors discuss the rationale for selection of 1 M LiPF₆ ethylene carbonate (EC)/diethyl carbonate (DEC) (1:1 by vol) + FEC (10%) + VC (1%). The electrolyte selection is almost as critical as the current collector, so this deserves some discussion.
- 5) It would be interesting if the authors are able to place all energy levels on the vacuum scale together with where the Li/Li⁺ redox sits in this liquid electrolyte. This could be done through calibration with a redox couple. See for example, J. Phys. Chem. Lett. 2014, 5, 14, 2419–2424.

Reviewer #2 (Remarks to the Author):

The authors report introducing a nano-window defect to carbon paper which lowers the Fermi level to reduce SEI formation due to electron tunneling. This is a thorough study that has many interesting results. However, there are many questions and concerns that must be resolved before publication.

1. Line 31: “While the efficacy of three dimensional (3D) porous current collectors in reducing the effective current density has been proven, their enlarged electroactive surface area can accompany a larger degree of electrolyte decomposition.

The authors highlight the effect of conventional 3D porous current collectors. Can the authors comment on how this effect is contributing to the improved performance of their d-CP current collector and how the benefits obtained from low Fermi-level surface introduced here can be distinguished from the reduction in current density obtained from the porosity of the current collector? The authors demonstrate how p-CP performs better than simple Cu—is that simply the benefit of the current collector porosity, and then then further benefit of the d-CP can be attributed to the NW defects?

The authors also highlight the issue of increased electrolyte decomposition due to the enlarged surface area of porous current collectors. The authors demonstrate how this is not a problem going from p-CP to d-CP which has a higher surface area. However, will this be a problem compared to conventional low surface area current collectors? For example, if the authors wanted to further improve the d-CP performance by using an optimized electrolyte, might this electrolyte degrade faster in a d-CP cell in comparison to a conventional Cu cell?

Furthermore, the authors show that electrolyte dry-out is a major issue for these cells when lean electrolyte conditions are used. Will the increased current collector porosity exacerbate this issue? Can dry-out be avoided with lean electrolytes using these porous current collectors?

2. Line 38: "The mosaic and grain-coalesced SEI formed on the current collector, which originates from aggressive electron tunneling from the current collector to the electrolyte, induces uneven Li nucleation, resulting in protrusion of the Li deposit through the defects in the SEI."..."however, we envisioned that lowering Fermi level at the surface of the current collector can enfeeble the electron tunneling by increasing the tunneling barrier."

Electron tunneling is surely a mechanism for SEI growth. However, it is not clear at all to this reviewer that the SEI formed as a result of electron tunneling is any worse than SEI formed for any other reason, i.e. direct contact with un-passivated lithium due to SEI breaking from large volume expansion and contraction. The entire thesis of this work is founded on the d-CP lowering the Fermi energy for mitigating SEI growth due to electron tunneling. Can the authors explain why avoiding SEI formed via electron tunneling is so important, especially since lithium inventory loss to the reformation of SEI due to large volume expansion and contraction seems more likely for lithium metal cells?

3. The authors should provide the porosity and pore volume of the carbon paper current collectors. The authors qualitatively demonstrate that lithium is plated in the interior volume of the d-CP with top-down and cross-sectional SEM images. It would be useful to know how much lithium could theoretically be accommodated in these current collectors.

4. The authors argue that the benefit of d-CP current collectors is the lowered Fermi level enabled by the NW defects which results in an optimized SEI. Can the authors comment how these NW defects may impact the performance of conventional Li-ion cells? For example, if these NW defects were introduced to conventional Li-ion cell-grade graphites to be used in a conventional Li-ion cell.

5. The authors demonstrate that the first cycle CE of the p-CP and d-CP are 80.8 and 87.4% respectively. These are quite low compared to decent graphites used in Li-ion cells that have first cycle CEs > 90%. Why is this the case? Does this speak to the issue of increased electrochemical surface area?

6. In Fig. 3a, the authors show the first cycle voltage curves of the p-CP and d-CP current collectors. The authors comment that the introduced defects of the d-CP does not significantly impact the delivered capacity. However, these data are plotted vs areal capacity (mAh/cm^2). Can the authors also include a specific capacity (mAh/g) scale so readers can easily tell how these capacities compare to normal graphites used in Li-ion cells? If it happens to be lower than the typical $\sim 350 \text{ mAh}/\text{g}$, can the authors explain why?

7. In the main text, there is no mention of the loading (mg/cm^2 and mAh/cm^2) of the carbon paper current collectors. It is very important to understand how much capacity is being intercalated into the graphite compared to how much is plated. This information must be included in the main text.

8. The authors do not comment in the main text how the d-CP current collector impacts volumetric energy density. Replacing an $8 \text{ }\mu\text{m}$ Cu current collector with $100 \text{ }\mu\text{m}$ d-CP will significantly decrease volumetric energy density. The authors should be forthright with this in the main text.

I also have some issues with the calculated energy densities presented in supplementary 3: The areal capacity used for these calculations is $4.2 \text{ mAh}/\text{cm}^2$, but the capacity demonstrated in Fig 7d is $< 4.0 \text{ mAh}/\text{cm}^2$. The average voltage used for all of these calculations is 3.9 V . However, the average voltage for a cell with some graphite will be lower than with a bare copper current collector (this is clearly shown in Fig 7b). The stack used to calculate these numbers only contains a single separator (shown in Fig S17a). However, a bi-cell stack requires two separators (two positive electrodes, two negative electrodes, two separators!). These discrepancies should be corrected.

Furthermore, the authors should include the total stack thickness they use to calculate the volumetric energy density in supplementary table 3. I do not get the same volumetric energy density numbers when I try to calculate it based on the thickness numbers provided in this table. There also seems to be a discrepancy between the provided volumetric energy density of Cu cells vs CP cells. The CP stack thickness is 50% greater than the Cu stack ($281 \text{ vs } 189 \text{ }\mu\text{m}$), but the energy density is only decreased to 80% ($1165 \text{ vs } 1418 \text{ Wh}/\text{L}$). If the stack is 1.5x thicker, the volumetric energy density should be 1.5x less.

9. Fig 6a presents coulombic efficiency on a scale from -10 to 110%. This is an unacceptable scale when CEs higher than 95% are required to even approach viability, and differences of just 1% make a very significant different to lifetime. The minimum CE on this scale should be 90%.

10. Throughout the paper, the authors claim their results have been achieved in an unpressurized system. On line 338 they state:

“It should be stressed that the cycle performance of d-CP in the AF-LMB is not derived from the pressure effect because stack pressure is not delivered to the inner surface of the 3D current collector.”

That may be so. However, it is my understanding that these results were achieved in a coin cell format. Unless the coin cells were constructed in some unconventional way which should be reported, then there is significant pressure exerted on the electrodes which will improve Li metal reversibility. Even if the authors above argument is true, there will be pressure exerted on the outer surfaces of the porous current collectors and Li plating there. Regardless, if the authors want to make the claim of achieving results in an unpressurized cell, they must present results that were obtained in a truly unpressurized cell (an unconstrained pouch, for example). The authors need to explain if these results are indeed from a conventional coin cell that has increased uniaxial pressure, and if so, remove or highly temper the claims of “unpressurized” results.

11. Line 58: “The defect-free SEI and Li-C orbital hybridization enable lateral Li deposition even in the internal structure of the 3D current collector where no external pressure is exerted normal on its surfaces”

I do not believe there is sufficient evidence to back up this claim. The authors provide top-down SEM images, but nothing that shows the morphology of the lithium in the internal porosity of the current collectors. The cross-sectional images seem to demonstrate that lithium indeed plates within the internal porosity, but it does not evidence that the lithium deposition inside the current collector is lateral. The authors seem to be inferring lateral deposition in the internal porosity from the lateral deposition demonstrated in Fig 5d-f which appears to be on the top surface of the current collector (which would be exposed to external pressure from the coin cell construction). I do not believe the authors have provided sufficient evidence to claim that they have enabled lateral Li deposition in the internal structure of the 3D current collector.

12. Fig 7d shows the lean electrolyte condition cycling up to 50 cycles. The d-CP show good capacity retention over this range, only 10% loss over 50 cycles (as reported in the abstract). However, Fig S16 shows that there is dramatic capacity roll over by cycle 60. I believe it is disingenuous for the full 100 cycle results to be hidden in the supplementary information. Fig 7c shows the flooded cells up to 100 cycles; the lean electrolyte cells should be shown to 100 cycles in the main paper as well. The challenges of lean electrolyte should not be minimized. The authors should be upfront about this and discuss these challenges in the main text.

13. In their conclusion, the authors state that “a rational design strategy for advancing AF-LMBs is presented.” Can the authors comment about how practical large scale synthesis of these current collectors might be? Without a copper substrate, will these carbon materials be flexible enough to be used in practical format cells (wound pouches or cylinders)? Also, it is well known that lithium electrodes thicken during aging due to the accumulation of dead lithium. With these current collectors be strong enough to accommodate this expansion or might they crumble due to larger internal volume expansion?

Response Letter

We thank for peer reviewing our manuscript and giving helpful and critical comments and advices. The constructive comments from the reviewer have significantly improved the quality of this manuscript. We have done further investigations about binding energies, diffusional behaviors of Li adatom on various surface and representation of Fermi level in main figures reflecting reviewers' advices. In addition, through deep contemplation of the reviewer's questions, we tried to clarify 1) the merits of a three-dimensional current collector with a low Fermi level that it has vaguely described in our previous manuscript and 2) that this current collector can have in a practical viewpoint. We hope that this revision can fulfill the reviewers' comments. The changes made are indicated by **yellow background** in the revised manuscript.

Title: Driving lean-electrolyte anode-free Li metal batteries by an electron-deficient three-dimensional current collector

Reviewer #1

1. The analysis shown in Fig. 5 is good, but the **reference state as Li bulk would be better as that automatically provides a voltage scale**. See for example, ACS Energy Lett., 2019, 4, 2952-2959.

Response: We appreciate for suggesting the paper that is closely related to our research. The binding energy shown in Figure.5I was obtained by subtracting the energy of each of the substrate and Li adatom from the energy of the structure in which Li adatom is adsorbed on the substrate, as written in Supplementary Note 1. All the calculated energies correspond to the total energy of each structure at 0 K, which is the same as gibbs free energy or enthalpy assuming that temperature contribution to free energy is negligible (ACS Energy Lett., 2019, 4, 2952-2959). Therefore, they are equivalent to the formula used to calculate nucleation overpotential presented in the suggested reference. In order to convey a clearer meaning, the term 'binding energy' was replaced with 'adsorption free energy, ΔG_{ads} ' in the revised manuscript, and the suggested reference (ref. 33 in the manuscript) was added to the revised manuscript.

2. What happens to the surface **diffusion barrier for these systems?** Do they observe any BEP relationship with the adsorption strength?

Fig. R1: Diffusional behavior of Li adatom on various substrates. a-d, Calculated Li adatom surface diffusion activation energies on (a) Li(111), (b) Li(110), (c) Li(200), (d) BLG. The blue atoms denote diffusing Li adatom.

e, Energy barrier for diffusion of Li adatom from inside to outside of NW-Li structure. **f**, Brønsted–Evans–Polanyi (BEP) relation between calculated adsorption enthalpy (ΔH_{abs}) and diffusion activation energy (E_{diff}) of Li adatom on various substrate.

Response: We are grateful for what the reviewer’s comment. As suggested, we calculated the diffusion barrier of lithium adatom on various surface structures. In the case of lithium’s dominant crystal plane ((200), (110), (111)) and BLG, the diffusion barrier was calculated through transition state (TS) search. In the case of the Li-NW structure, it was difficult to define a diffusion path, so the energy difference of inside and outside of defect (Fig. 5l) was considered as activation barrier for diffusing from inside of defect to outside. As the reviewer commented, our calculation results showed a linear relationship between adsorption enthalpy and diffusion barrier.

Through BEP relationship analysis, it was clarified that the current collector with strong binding energy can suppress the surface diffusion of lithium adatom on the surface. Owing to the result, we are able to more clearly deliver our claim; contrary to the rapid surface Li diffusion and consequent formation of non-uniform Li cluster on the graphene surface, the NW-Li structure does not bring coalesce of Li nuclei due to the limited surface Li diffusion.

These new data would be meaningful to potential readers of our manuscript, thus were added the data and reference to the revised manuscript.

Revised manuscript:

(Supplementary Fig. 15)

Supplementary Figure 15. Diffusional behavior of Li adatom on various substrates. a-e, Calculated Li adatom surface diffusion activation energies on (a) Li(111), (b) Li(110), (c) Li(200), (d) BLG, and (e) NW-Li. The blue atoms denote diffusing Li adatom. **f**, Brønsted–Evans–Polanyi (BEP) relation¹⁴ between calculated adsorption enthalpy and diffusion activation energy of Li adatom on various substrate.

3. In Fig 7, the authors should compare against a few other data points from PNAS 2020 117 (44) 27195-27203 and the associated data points there.

Response: The performance comparison in Fig. 7 was intended to make fair comparison among AF-LMBs. The suggested reference deals with lithium metal battery. Thus, it may be not appropriate to include the suggested data points in Fig. 7. We hope the reviewer understand this point. The suggested reference is a highly valuable reference for delivering the current state of art. In this aspect, it was referenced in the introduction section of the revised manuscript (ref. 15 in the manuscript).

4. Can the authors discuss the rationale for selection of 1 M LiPF₆ ethylene carbonate (EC)/diethyl carbonate (DEC) (1:1 by vol) + FEC (10%) + VC (1%). The electrolyte selection is almost as critical as the current collector, so this deserves some discussion.

Response: We highly appreciated the reviewer's comments for our manuscript and strongly agree that it is worth discussing about the rationality of electrolyte selection. In our anode-free battery, since lithiation/de-lithiation of graphite as well as plating/stripping of lithium occurs, carbonate solvents (EC, DEC and FEC) were selected to prevent exfoliation of graphite rather than etheral solvents (Adv. Energy Mater. 2017, 7, 1700418). For the other reason, we targeted to employ high capacity NCM electrode; the high voltage cathode deters etheral electrolytes but prefers carbonate electrolytes. The LiPF₆ salt was chosen to only focus on the performance change depending on the current collector when using the most commonly used lithium salt. 2% of VC was added to stabilize initial surface film (SEI and CEI) structure on anode and cathode (J. Electrochem. Soc. 2009, 156, A103-A113). We added the discussion to the revised manuscript.

Revised manuscript:

(Line 357-362)

In our anode-free battery, since lithiation/de-lithiation of graphite as well as plating/stripping of lithium occurs, carbonate solvents (EC, DEC and FEC) were selected to prevent exfoliation of graphite rather than etheral solvents³⁶. In addition, the carbonate electrolytes enable the use of NCM cathode due to their high oxidation stability. LiPF₆ was chosen due to its high ionic conductivity in carbonate solvents. 2% of VC was added to help the formation of stable SEI and cathode-electrolyte interphase (CEI)³⁷.

5. It would be interesting if the authors are able to place all energy levels on the vacuum scale together with where the Li/Li⁺ redox sits in this liquid electrolyte. This could be done through calibration with a redox couple. See for example, J. Phys. Chem. Lett. 2014, 5, 14, 2419–2424.

Fig. R2: Experimental and simulated Fermi-level landscape during lithiation. **a**, Plots of Fermi level for p-CP and d-CP during lithiation from OCV to 1.0 V (V vs. Li/Li⁺) determined from extrapolation of the secondary cut-off region in UPS. All the Fermi levels of Li/Li⁺, redox associated with electrolyte reduction, Li metal, p-CP and d-CP were calibrated with vacuum state of $E_{vac}=0$ eV. **b**, Calculated surface Fermi-level and atomic structure of BLG (green plots), NW-Li (red plots) and calculated redox potential of lithiation of NW (blue plots).

Response: We thank for the reviewer's detailed comments to improve the quality of our manuscript. Since electrode Fermi level is a critical factor to the redox reactions of Li/Li⁺ and electrolyte decomposition, displaying redox potentials and Fermi level of electrode surface on vacuum level scale altogether may increase reader's understanding. As the reviewer suggested, we revised the figure and added the provided reference (ref. 27 in the manuscript).

In addition, to support the UPS result, we newly calculated Fermi levels of lithiated BLGs at different degree of lithiation and added them to new figure. In line with the UPS result, the NW structure has a lower Fermi level than BLG at all the lithiation degrees. Based on the additional results, we revised Fig. 4 and related main text.

Revised manuscript:

(Figure 4)

Fig. 4: Atomistic and electronic structure of NW defect during lithiation. **a**, Voltage profiles for the early stage of galvanostatic lithiation for p-CP and d-CP at 0.1 mA cm⁻² in half-cell configuration. **b, c**, Image plots of operando XRD spectra along with the potential profile during the first lithiation and deconvoluted XRD patterns at OCV, 1.5 V and 0.8 V for p-CP (b) and d-CP (c). The peaks from the lithiated defect structure (lithiated NW, Li₃C₈) at 43° of 2θ are indicated by blue lines. **d**, Plots of Fermi

level for p-CP and d-CP during lithiation from OCV to 1.0 V (V vs. Li/Li⁺) determined from extrapolation of the secondary cut-off region in UPS. All the Fermi levels of Li/Li⁺, redox associated with electrolyte reduction, Li metal, p-CP and d-CP were calibrated with vacuum state of $E_{vac}=0$ eV. e, The relaxed structures of lithiated BLG, NW and fully lithiated NW (Li₃C₈), corresponding DFT calculations of Fermi level and redox potential for NW at various lithiation degrees. The Li, carbon at the upper layer, and carbon at the lower layer are shown in blue, yellow, and gray color, respectively.

(Line 177-213)

The atomic structural changes of p-CP and d-CP during the early stage lithiation were traced by an operando X-ray diffraction (XRD) analysis (Fig. 4b, c). The XRD image plots in the 24-30° region shed light on the lithiation process of the graphitic structure of carbon fiber. In contrast to the strong peak at 2θ of 26.5° from G(002) of p-CP, the weak intensities at around 26.5° for d-CP indicate the existence of a defective carbon layer. For d-CP, the lithiated structures of LiC₆ and LiC₁₂ started to appear at lower lithiation capacities than for p-CP, suggesting that relatively fast Li transport through the defective layer facilitates the lithiation of the inner graphitic carbon fiber. The structural evolution of the defect structure can be traced from the higher 2θ region. From the beginning of the lithiation, a peak at $2\theta=43^\circ$ appeared for d-CP and gradually grew with decreasing potential, indicating Li insertion into the NW defect structure of the graphite surface^{24,25}. By contrast, the XRD pattern of p-CP was invariant in the potential range where solid-solution type lithiation at the 1L stage of initial graphite lithiation (Li_xC₆, $x<0.16$)²⁶ occurs; the solid-solution type lithiation does not change the lattice structure of graphite surface.

In order to trace the electronic structure along with the variation of surface atomic structure, the work function, which is the difference between the vacuum level (E_{vac}) and Fermi level²⁷, was measured along the potential descending for p-CP and d-CP by using UV photoelectron spectroscopy (UPS, Supplementary Fig. 4). The work function of the intrinsic electrode surface of p-CP is 4.52 eV, which is coincident with the work function of the graphite reported in previous studies²⁸, and that of d-CP is 4.80 eV (Fig. 4d). Fermi levels of p-CP and d-CP increased with decreasing potential upon lithiation. Notably, Fermi levels of d-CP were much lower than those of p-CP regardless of the degree of lithiation, indicating that the surface of d-CP has a lower electron energy level upon charging.

In line with the UPS result, DFT calculation also demonstrated that the NW structure has a lower Fermi level than BLG during lithiation. Based on the 2D/G ratio in raman spectroscopy (Fig. 2h) and pore size distribution (Fig. 2i), we constructed bilayer graphene (BLG) and NW defect in BLG (Supplementary Fig. 5, 6). According to the calculations of the sequential lithiation of NW, a Li atom spontaneously penetrates through the NW hole, accepting electrons of Li by the electron-deficient NW (Supplementary Fig. 7, 8) and places between the graphene layers with a gradual diminution of spin momentum (Supplementary Fig. 9, 10) until a stoichiometry of Li₃C₈ (fully lithiated state) is reached. The calculated potentials profile and Fermi levels of Li-NW structures at different degrees of lithiation are shown in Fig. 4e. The potential change with lithiation accounts for the larger lithiation capacity of d-CP above 1.5 V vs. Li/Li⁺. In accordance with the UPS results, the calculated Fermi levels of the NW structure were much lower than those of BLG. The results indicate that the Fermi level of d-CP is lower than that of p-CP even though the NW was lithiated, and the electron-deficient properties of d-CP caused a change in the surface SEI structure.

Reviewer #2

1. Line 31: “While the efficacy of three dimensional (3D) porous current collectors in reducing the effective current density has been proven, their enlarged electroactive surface area can accompany a larger degree of electrolyte decomposition.

The authors highlight the effect of conventional 3D porous current collectors. Can the authors comment on how this effect is contributing to the improved performance of their d-CP current collector and how the benefits obtained from low Fermi-level surface introduced here can be distinguished from the reduction in current density obtained from the porosity of the current collector? The authors demonstrate how p-CP performs better than simple Cu—is that simply the benefit of the current collector porosity, and then then further benefit of the d-CP can be attributed to the NW defects?

The authors also highlight the issue of increased electrolyte decomposition due to the enlarged surface area of porous current collectors. The authors demonstrate how this is not a problem going from p-CP to d-CP which has a higher surface area. However, will this be a problem compared to conventional low surface area current collectors? For example, if the authors wanted to further improve the d-CP performance by using an optimized electrolyte, might this electrolyte degrade faster in a d-CP cell in comparison to a conventional Cu cell?

Furthermore, the authors show that electrolyte dry-out is a major issue for these cells when lean electrolyte conditions are used. Will the increased current collector porosity exacerbate this issue? Can dry-out be avoided with lean electrolytes using these porous current collectors?

Fig. R3: a, EIS analysis for the half-cells with copper, p-CP and d-CP after the pre-cycle. b, c, Quantification of irreversible capacity caused by Li metal plating/stripping (b) or sequential restoration of damaged SEI on current collector surface (c). The restoration of SEI was performed by galvanostatic cycling at a current density of 0.1 mA cm⁻² in the range of 0 to 2.0 V.

Fig. R4: a-c, SEM images after 10 cycles of 5 mAh cm⁻² Li deposition at 2 mA cm⁻² for Cu (a), p-CP (b) and d-CP (c). Electrolyte: 1 M LiPF₆ EC/DEC 1:1 v/v 10% FEC 1% VC

Response:

The authors highlight the effect of conventional 3D porous current collectors. Can the authors comment on how this effect is contributing to the improved performance of their d-CP current collector and how the benefits obtained from low Fermi-level surface introduced here can be distinguished from the reduction in current density obtained from the porosity of the current collector? The authors demonstrate how p-CP performs better than simple Cu—is that simply the benefit of the current collector porosity, and then then further benefit of the d-CP can be attributed to the NW defects?

We appreciate the reviewer's constructive question. In the previous our manuscript as the reviewer said, we simply explain that the reason why p-CP shows better performance than Cu is because of its three-dimensional structure. Complete decoupling of the current density effect and low-Fermi level effect is highly challenging. However, we can assure that the good performance of d-CP is not due to the current density effect. It is demonstrated by the comparison of impedance and irreversible capacity loss between Cu and p-CP. As shown in Fig. R3a, the interfacial resistance of p-CP is much smaller than that of Cu, indicating that effective surface area is a dominating factor for interfacial resistance as reported in a previous literature (*Adv. Funct. Mater.* 2017, 27, 1606422). However, the coulombic efficiency for Li deposition and dissolution was quite similar between p-CP (89.2%) and Cu (90.2%) as shown in Fig. R3b, which imply that the irreversible capacity due to the large surface area offset the current density effect. It is supported by the subsequent cycle in the range of 0-2.0 V. The irreversible capacity found in the subsequent cycle comes from the restoration of SEI by electrolyte decomposition. The Cu and p-CP showed 0 and 0.143 mAh cm⁻² of irreversible capacity, verifying intensive electrolyte decomposition on 3D current collector. Therefore, the benefit of the current collector porosity cannot be the reason for the outstanding performance of d-CP.

The authors also highlight the issue of increased electrolyte decomposition due to the enlarged surface area of porous current collectors. The authors demonstrate how this is not a problem going from p-CP to d-CP which has a higher surface area. However, will this be a problem compared to conventional low surface area current collectors? For example, if the authors wanted to further improve the d-CP performance by using an optimized electrolyte, might this electrolyte degrade faster in a d-CP cell in comparison to a conventional Cu cell?

We appreciate the reviewer for the valuable questions. Because the anode-free electrode is a complex system, including electrolyte decomposition effect at not only current collector but also Li deposit and geometry effect, it is hard for us to make a concrete answer to the reviewer's question. However, we would like to deliver your insights on the interplay among the various effects in the following.

As the reviewer commented, the electrolyte decomposition at low surface area current collector surface is not as serious as 3D current collector. However, as observed for Cu current collector, its reversibility of Li plating/stripping is poor due to the inhomogeneous Li plating. It means that, for conventional low surface area current collector, the issue of Li deposition morphology and consequent electrolyte decomposition at Li deposit would be more important than the amount of electrolyte decomposition at Cu surface.

Although the irreversible capacity (Li plating/stripping) for d-CP is much smaller than that for Cu (Fig. R3b), there are more electrolyte decomposition on d-CP surface due to the larger surface area (Fig. R3c). If the electrolyte decomposition at current collector surface can be mitigated by using an optimized electrolyte as exemplified by the reviewer, the advantage of 3D current collector (reduction of effective current density) can be also pronounced, possibly resulting in a better performance.

Recent electrolyte optimization is directed to form a stable SEI by means of electrolyte decomposition (*Nat. Energy* 2021, doi: 10.1038/s41560-021-00787-9). Therefore, gradual electrolyte consumption is inevitable. However, in the case of d-CP, Li morphology can be controlled by surface electronic property with minimizing the electrolyte consumption. In this aspect, we expect that electrolyte optimization and low-Fermi level surface can be complementary; the optimized electrolyte forms a stable SEI and low-Fermi level surface minimize its expense.

Furthermore, the authors show that electrolyte dry-out is a major issue for these cells when lean electrolyte conditions are used. Will the increased current collector porosity exacerbate this issue? Can dry-out be avoided with lean electrolytes using these porous current collectors?

The electrolyte dry out is influenced not only the geometry of current collector but also roughness of Li deposit and dead-Li layer. Even though current collector porosity exacerbates this issue, it can be offset or

overcompensated by minimizing electrolyte decomposition on current collector and inducing uniform Li deposit as demonstrated with d-CP in this work. In contrast to d-CP, Cu and p-CP form a thick dead-Li layer after cycling (Fig. R4), aggregating the issue of electrolyte dry out.

After contemplating the reviewer's comments, we added the data corresponding to Cu (Fig. R3b and c) was added to Fig. 6j and k to describe the comparison with the 2D current collector in more detail.

Revised manuscript:

(Figure 6)

Fig 6: Reversibility of Li plating/stripping on the defective carbon surface. **a**, The plots of CE as a function of cycle number for Cu, p-CP, and d-CP current collector in half-cell configuration at a current density of 2.0 mA cm⁻² and a capacity of 5.0 mAh cm⁻². **b-e**, Li deposition morphologies for p-CP (b-e) and d-CP (f-i) at various plating capacities from 3.0 to 9.0 mAh cm⁻². **j, k**, Quantification of the irreversible capacity caused by Li metal plating/stripping and sequential restoration of damaged SEI on current collector surface. The restoration of SEI was performed by a galvanostatic cycling in the range of 0.0 and 2.0 V at a current density of 0.1 mA cm⁻². **l, m**, Schematics for the irreversible loss of Li and the electrolyte on p-CP surface (l), and the proposed mechanism for prevention of the irreversibility on d-CP surface (m).

(Line 331-337)

The irreversible capacity of the first cycling originates from the formation of SEI on current collector and deposited Li and of isolated Li, whereas that of the second partial cycling should reflect the restoration of the SEI on the current collector damaged during the first cycle. The capacity loss from the Li plating/stripping was 0.490 mAh cm⁻² (CE of 90.2 %) for Cu and 0.541 mAh cm⁻² (CE of 89.2 %) for p-CP (Fig. 6j) which points out that larger amount of Li was lost with p-CP and that the SEI formation on its large surface area offsets the positive effect of a reduced current density.

2. Line 38: “The mosaic and grain-coalesced SEI formed on the current collector, which originates from aggressive electron tunneling from the current collector to the electrolyte, induces uneven Li nucleation, resulting in protrusion of the Li deposit through the defects in the SEI.”...”however, we envisioned that lowering Fermi level at the surface of the current collector can enfeeble the electron tunneling by increasing the tunneling barrier.”

Electron tunneling is surely a mechanism for SEI growth. However, it is not clear at all to this reviewer that the SEI formed as a result of electron tunneling is any worse than SEI formed for any other reason, i.e. direct contact with un-passivated lithium due to SEI breaking from large volume expansion and contraction. The entire thesis of this work is founded on the d-CP lowering the Fermi energy for mitigating SEI growth due to electron tunneling. Can the authors explain why avoiding SEI formed via electron tunneling is so important, especially since lithium inventory loss to the reformation of SEI due to large volume expansion and contraction seems more likely for lithium metal cells?

Response: We appreciated the reviewer for critical comments for helping us improve logic of this manuscript. In anode-free lithium batteries, we agree with the reviewer's comment that dead-Li and a continuous reaction of lithium and electrolyte are directly connected to capacity deterioration. As the reviewer commented, according to Figure 6j and k, the irreversible capacity caused by dead-Li and the side reaction is larger than that formed by electron tunneling on the current collector before Li deposition. However, we would like to emphasize that the uniformity of SEI layer on the current collector formed by electron tunneling from the current collector to the electrolyte not only influences the SEI breakage and reformation, but also subsequent uniform initial Li nucleation and growth. The uniform flux of Li⁺ thorough the uniform SEI on current collector is a prerequisite for uniform Li nucleation. As the uniformity of Li deposit is increased, the Li inventory loss is decreased due to a reduced interfacial stress. We agree that this part was ambiguous in the previous manuscript, thus revised to be clearer.

Revised manuscript:

(Line 35-41)

The properties of the mosaic and grain-coalesced SEI formed on the current collector surface, which originate from aggressive electron tunneling from the current collector to the electrolyte, can induces continuous electrolyte decomposition and defective SEI and thereby uneven initial Li nucleation and growth. In this regard, we envisioned that it is possible to modify the microstructure of SEI and the behavior of Li nucleation/growth via tuning Fermi level of the current collector surface.

3. The authors should provide the porosity and pore volume of the carbon paper current collectors. The authors qualitatively demonstrate that lithium is plated in the interior volume of the d-CP with top-down and cross-sectional SEM images. It would be useful to know how much lithium could theoretically be accommodated in these current collectors.

Electrodes	Apparent density (g cm ⁻³)	Carbon fiber density (g cm ⁻³)	Porosity	Pore volume (cm ³ per 1 cm ²)	Maximum Li accommodation (mAh cm ⁻²)
p-CP	0.55	2.26	0.757	7.57x10 ⁻³	15.6
d-CP	0.39	2.26	0.827	8.27x10 ⁻³	17.1

Table R1: Calculated electrode porosity and the maximum capacity of lithium metal that can be stored in the pores of the electrode.

Fig. R5: a, b, SEM images of p-CP (a), d-CP (b) after Li deposition at 0.5 mA cm^{-2} and 16 mAh cm^{-2}

Response: We thank the reviewer for the comments. We calculated the porosity, pore volume, and maximum Li accommodation for p-CP and d-CP from the apparent density of p-CP and d-CP, carbon fiber density, and Li metal density, and listed them in Table R1. We added the data to revised manuscript and supplementary information.

Revised manuscript:

(Line 326-328)

Unlike p-CP, top plating was not observed for d-CP even at the Li deposition capacity (16 mAh cm^{-2}) which corresponds to full accommodation of its pores (Supplementary Table 3 and Supplementary Fig. 18).

(Supplementary Information)

Electrodes	Apparent density (g cm^{-3})	Carbon fiber density (g cm^{-3})	Porosity	Pore volume ($\text{cm}^3 \text{ per } 1 \text{ cm}^2$)	Maximum Li accommodation (mAh cm^{-2})
p-CP	0.55	2.26	0.757-	7.57×10^{-3}	15.6
d-CP	0.39	2.26	0.827	8.27×10^{-3}	17.1

Supplementary Table 3. Calculated electrode porosity and the maximum capacity of lithium metal storage in the pores of the electrodes.

Supplementary Figure 18. a, b) SEM images of a) p-CP , b) d-CP after Li deposition at 0.5 mA cm^{-2} and 16 mAh cm^{-2}

4. The authors argue that the benefit of d-CP current collectors is the lowered Fermi level enabled by the NW defects which results in an optimized SEI. Can the authors comment how these NW defects may impact the performance of conventional Li-ion cells? For example, if these NW defects were introduced to conventional Li-ion cell-grade graphites to be used in a conventional Li-ion cell.

Fig. R6: Coulombic efficiency test of carbon current collector||Li half cell

Response: We appreciate the reviewers' wide view. To check the possibility of improving efficiency for intercalation chemistry with NW, we investigated the reversibility of Li||p-CP and Li||d-CP half-cell with a 1 M LiPF₆ EC/DEC (1:1 v/v). As shown in Fig. R6, the intercalation/de-intercalation efficiency of d-CP was nearly 100%, while p-CP electrode showed a lower CE of 99.68%. The result suggests that the uniform SEI structure derived by NW defects also works for Li-ion batteries.

Revised manuscript:

(Line 136-137)

As a graphite electrode, d-CP showed higher CEs than p-CP for Li intercalation/de-intercalation process, verifying the lowered electrolyte decomposition (Supplementary Fig. 3).

(Supplementary information)

Supplementary Figure 3. Coulombic efficiency test of carbon current collector||Li half cell

5. The authors demonstrate that the first cycle CE of the p-CP and d-CP are 80.8 and 87.4% respectively. These are quite low compared to decent graphities used in Li-ion cells that have first cycle CEs > 90%. Why is this the case? Does this speak to the issue of increased electrochemical surface area?

Response: We are grateful for what the reviewer comments. It is true that carbon paper (p-CP) used in our research has a relatively lower ICE value than the of a graphite electrode (~85%, Nano energy 2020, 77, 105143). However, when calculating the irreversible specific capacity, p-CP is about 43.6 mAh g⁻¹ and d-

CP is 40.4 mAh g^{-1} , which is similar to the first cycle irreversible capacity reported in other papers ($40\text{--}100 \text{ mAh g}^{-1}$, Solid State Ionics 2005, 176, 905-909). In the case of carbon paper, since the reversible capacity is smaller than that of typical graphite (details described in question 6), the ICE value can be lower in spite of similar irreversible capacity.

6. In Fig. 3a, the authors show the first cycle voltage curves of the p-CP and d-CP current collectors. The authors comment that the introduced defects of the d-CP does not significantly impact the delivered capacity. However, these data are plotted vs areal capacity (mAh/cm^2). Can the authors also include a specific capacity (mAh/g) scale so readers can easily tell how these capacities compare to normal graphites used in Li-ion cells? If it happens to be lower than the typical $\sim 350 \text{ mAh/g}$, can the authors explain why?

Fig. R7: Specific capacity of Li intercalation into carbon paper. a, p-CP b, d-CP.

Response: We appreciate the comments and agree with the reviewer's proposal. We measured the specific capacity according to the current density (0.1 mA cm^{-2} , 0.5 mA cm^{-2} , 2.0 mA cm^{-2}) because the intercalation capacity of carbon paper differs with current density. Carbon paper exposes basal plane more than edge plane, so its specific capacity can be smaller than 300 mAh g^{-1} (Nat. Commun. 2017, 8, 812). The charge/discharge voltage profile for p-CP and d-CP was shown in Fig. R7a and b, respectively. The measured specific capacity was 165, 119 and 55.2 mAh g^{-1} for p-CP and 245, 164 and 83.4 mAh g^{-1} for d-CP at 0.1, 0.5 and 2.0 mA cm^{-2} , respectively. d-CP has higher specific capacities due to a promoted Li insertion through the defect hole (Line 193-195). These data were added to the supplementary information of the revised manuscript.

Revised manuscript:

Supplementary Figure 19. Li intercalation and deintercalation potential profiles at various current densities for a, p-CP b, d-CP

According to the half-cell tests (Supplementary Fig. 19) and areal loading (Supplementary Table 3), the areal intercalation capacity is 0.908, 0.654 and 0.304 mAh cm⁻² for p-CP and 0.956, 0.651 and 0.322 mAh cm⁻² for d-CP at 0.1, 0.5, and 2.0 mA cm⁻², respectively. The Li plating capacity for 2.0 mA cm⁻² and 4 mA cm⁻² in anode-free cell operation is 3.70 mAh cm⁻² for p-CP and 3.68 mAh cm⁻² for d-CP.

7. In the main text, there is no mention of the loading (mg/cm² and mAh/cm²) of the carbon paper current collectors. It is very important to understand how much capacity is being intercalated into the graphite compared to how much is plated. This information must be included in the main text.

Response: We appreciate the reviewer's detailed comment on intercalation capacity and deposition capacity. According to the half cell tests (Fig. R7), the areal intercalation capacity is 0.908, 0.654 and 0.304 mAh cm⁻² for p-CP and 0.956, 0.651 and 0.322 mAh cm⁻² for d-CP at 0.1, 0.5, and 2.0 mA cm⁻², respectively. The Li plating capacity for 2.0 mA cm⁻² and 4 mA cm⁻² in anode-free cell operation is 3.70 mAh cm⁻² for p-CP and 3.68 mAh cm⁻² for d-CP. Because areal intercalation capacity (mAh cm⁻²) is important when using carbon as an anode current collector, as suggested by the reviewer, the areal capacity of p-CP and d-CP were included in the main text with a proper supplementary figure.

Revised manuscript:

(Line 368-371)

Figure 7c shows the cycling performance of the anode-free cells at practical charging and discharging current density of 2 mA cm⁻² (0.48 C) under the flooded and lean electrolyte condition. The Li plating capacity and intercalation capacity for the operation were 3.70 and 0.30 mAh cm⁻² for p-CP, and 3.68 and 0.32 mAh cm⁻² for d-CP (Supplementary Fig. 19).

8. The authors do not comment in the main text how the d-CP current collector impacts volumetric energy density. Replacing an 8 um Cu current collector with 100 um d-CP will significantly decrease volumetric energy density. The authors should be forthright with this in the main text.

I also have some issues with the calculated energy densities presented in supplementary 3: The areal capacity used for these calculations is 4.2 mAh/cm², but the capacity demonstrated in Fig 7d is < 4.0 mAh/cm². The average voltage used for all of these calculations is 3.9 V. However, the average voltage for a cell with some graphite will be lower than with a bare copper current collector (this is clearly shown in Fig 7b). The stack used to calculate these numbers only contains a single separator (shown in Fig S17a). However, a bi-cell stack requires two separators (two positive electrodes, two negative electrodes, two separators!). These discrepancies should be corrected.

Furthermore, the authors should include the total stack thickness they use to calculate the volumetric energy density in supplementary table 3. I do not get the same volumetric energy density numbers when I try to calculate it based on the thickness numbers provided in this table. There also seems to be a discrepancy between the provided volumetric energy density of Cu cells vs CP cells. The CP stack thickness is 50% greater than the Cu stack (281 vs 189 um), but the energy density is only decreased to 80% (1165 vs 1418 Wh/L). If the stack is 1.5x thicker, the volumetric energy density should be 1.5x less.

Response: We are very grateful for the reviewer's critical comment and reviewing energy density calculation described in our manuscript. As the reviewer commented, the volumetric energy density of discharged anode-free cell is higher for copper current collector than carbon current collector. However, for Cu current collector, the volumetric energy density of charged state is lower than that of discharged state, which is contrasted by indifference for 3D current collector. Furthermore, ramified Li deposition and build up of porous layer on Cu current collector diminish the volumetric energy density of charged cell upon cycling. Therefore, for fair comparison, the volumetric energy density was calculated using the

thickness of negative electrode at fully charged state as reported in previous literature (Nat. Energy 2020, 5, 693-702).

Assuming dendrite-free Li deposition on Cu current collector, the anode thickness at fully charged state (4.2 mAh cm⁻² at 0.1 C, and 5 μm per 1 mAh cm⁻²) was 50 μm (8 μm (Cu) + 21 μm (Li)*2 (bi cell)) for the copper anode. In the case of p-CP and d-CP, the anode thickness was calculated as 100 μm assuming that 4.2 mAh cm⁻² of lithium was electrodeposited inside the carbon structure. In the case of d-CP, this is true as shown in Supplementary Fig. 14l, but in the case of p-CP, as shown in Supplementary Fig. 14k, a slight thickening of the negative electrode actually occurred during the charging process. The calculated volumetric densities were stated in the main text of the revised manuscript.

Following the indication by reviewer, discharge capacity and average voltage were re-measured at 0.1 C thoroughly; the average voltage of Cu, p-CP and d-CP were 3.860, 3.844, and 3.850 V. The number of PP separator for calculating the energy densities presented in Supplementary Table 3 was correct as two, but the scheme of Supplementary Fig. 17a and relevant description were incorrect. We added a 0.1 C charge/discharge profile (Fig. 7a, revised figure is provided in below question 12), description on how energy density is calculated, the modified the scheme (Supplementary Fig. 21a), and corrected supplementary table 4 in the revised manuscript.

Revised manuscript:

(Line 362-364)

The voltage profiles of the anode-free cell with p-CP or d-CP (Fig. 7a) showed a distinctive carbon lithiation step at the beginning of charging (shaded in blue), as observed for the half cells.

(Line 386-395)

The cell-level energy density of AF-LMBs with different current collector was estimated based on the voltage profiles shown in Fig. 7a. The AF-LMB with d-CP can deliver energy densities of 349.1 Wh kg⁻¹ and 1084.7 Wh l⁻¹. Assuming void-free Li deposition (5 μm per 1.0 mAh cm⁻² Li), the cell with the Cu current collector has a lower gravimetric energy density (338.2 Wh kg⁻¹) but a higher volumetric energy density (1304 Wh l⁻¹) compared with the d-CP cell (Supplementary Table 4, Supplementary Fig. 21). However, when the increased electrode thickness due to the Li dendritic growth and the formation of a thick porous layer is reflected in the calculation, the volumetric energy density of the Cu cell after 10 cycle is lowered to 790.8 Wh l⁻¹, which is contrasted by the invariant energy density for d-CP (Supplementary Fig. 22).

(Supplementary Information)

	Cu	p-CP	d-CP
Al current collector (13 μm)	3.51 mg cm ⁻²	3.51 mg cm ⁻²	3.51 mg cm ⁻²
Anode current collector	7.17 mg cm ⁻² (8 μm)	5.5 mg cm ⁻² (100 μm)	3.9 mg cm ⁻² (100 μm)
NMC811 active material (74 μm)	20.79 mg cm ⁻²	20.79 mg cm ⁻²	20.79 mg cm ⁻²
PP separator (20 μm)	1.95 mg cm ⁻²	1.95 mg cm ⁻²	1.95 mg cm ⁻²
Electrolyte	E/C=4.84 g Ah ⁻¹	E/C=4.84 g Ah ⁻¹	E/C=4.84 g Ah ⁻¹
0.1 C discharge capacity	4.24 mAh cm ⁻²	4.14 mAh cm ⁻²	4.24 mAh cm ⁻²
Average voltage	3.860 V	3.844 V	3.850 V
Total stack energy	32.74 mWh cm ⁻²	31.83 mWh cm ⁻²	32.65 mWh cm ⁻²
Total stack weight	96.8 mg cm ⁻²	93.5 mg cm ⁻²	93.5 mg cm ⁻²
Total stack thickness (Charged)	251 μm (ideally)	301 μm (ideally)	301 μm
Stack gravimetric energy density	338.2 Wh kg⁻¹	340.2 Wh kg⁻¹	349.1 Wh kg⁻¹

Stack volumetric energy density	1304.4 Wh l ⁻¹	1057.1 Wh l ⁻¹	1084.7 Wh l ⁻¹
80% capacity retention cycle #	4 cycles	4 cycles	50 cycles

Supplementary Table 3. Estimated gravimetric and volumetric energy densities of AF-LMB cells with different current collector.

Supplementary Figure 21. Gravimetric energy density and cycle life of AF-LMB cells. a) Schematic illustration of a stacked AF-LMB. **b)** Estimated gravimetric energy densities of the AF-LMB with different current collectors. **c)** Number of cycles at 80% capacity retention for the AF-LMBs with different current collectors.

The gravimetric energy densities of a stacked AF-LMB with different current collector were calculated based on 0.1 C charge/discharge voltage profile (Fig. 7a). In this calculation, the single bi-cell constitutes one sheet of double-side coated NCM cathode, one sheet of fully charged anode (current collector + deposited Li), two sheet of separator and electrolyte²⁴. Here, in the case of Cu or p-CP, the thickness of the negative electrode increases significantly as Li deposition, but in the case of Cu, it is assumed that 5 μm per 1 mAh cm⁻², and in the case of p-CP, lithium is well deposited into the pore.

Supplementary Fig. 22. a-c, SEM images fully discharged state of anode current collectors after 10 cycles of 5 mAh cm⁻² Li plating/stripping at 2 mA cm⁻² for Cu (a), p-CP (b) and d-CP (c). Electrolyte: 1 M LiPF₆ EC/DEC 1:1 v/v 10% FEC 1% VC

In the case of the Cu current collector, it can be seen that a porous layer of about 100 μm is formed in only 10 cycles. When a bi-cell structured cell is constructed, the thickness of the single stack will be increased by 200 μm (100 μm each side), resulting in almost twice the volumetric energy density reduction (thickness vary from 251 at μm fully charged state to 413 μm at fully discharged state after 10 cycles), resulting volumetric energy density of 790.8 Wh l⁻¹. On the other hand, in the case of d-CP, it was confirmed that the thickness of the current collector was almost constant even after cycling.

9. Fig 6a presents coulombic efficiency on a scale from -10 to 110%. This is an unacceptable scale when CEs higher than 95% are required to even approach viability, and differences of just 1% make a very significant different to lifetime. The minimum CE on this scale should be 90%.

Response: We appreciate the reviewer's comment. We agree with the reviewer that it is difficult to identify a CE difference of 1% on the current CEs scale. However, in the case of Cu and p-CP, since the CE is close to 90%, the data may be obscured, so the y-axis scale is set to 60%-105%, and then the average CE for each sample is indicated through the dotted line. As the reviewer suggested, Figure 6 was revised.

10. Throughout the paper, the authors claim their results have been achieved in an unpressurized system. On line 338 they state:

“It should be stressed that the cycle performance of d-CP in the AF-LMB is not derived from the pressure effect because stack pressure is not delivered to the inner surface of the 3D current collector.”

That may be so. However, it is my understanding that these results were achieved in a coin cell format. Unless the coin cells were constructed in some unconventional way which should be reported, then there is significant pressure exerted on the electrodes which will improve Li metal reversibility. Even if the authors above argument is true, there will be pressure exerted on the outer surfaces of the porous current collectors and Li plating there. Regardless, if the authors want to make the claim of achieving results in an unpressurized cell, they must present results that were obtained in a truly unpressurized cell (an unconstrained pouch, for example). The authors need to explain if these results are indeed from a conventional coin cell that has increased uniaxial pressure, and if so, remove or highly temper the claims of “unpressurized” results.

Response: We appreciate the reviewer for the comment. As in the reviewer's comment, pressure can be exerted on the electrode in the coin cell and it is not clear whether the pressure is delivered to the inner surface of 3D-current collector. To avoid confusion, so we removed the term ‘unpressurized’ and the related claims in the text.

Revised manuscript:

(Line 29-32)

While the efficacy of three dimensional (3D) porous current collectors in reducing the effective current density has been proven, their enlarged electroactive surface area can accompany a larger degree of electrolyte decomposition at the current collector-electrolyte interface¹².

(Line 57-60)

We report a successful introduction of 3D current collector in an NCM-based anode-free Li metal battery, that shows capacity retention of 90% at 50 cycles at high current density (2.0 mA cm⁻²), high areal capacity (4.2 mAh cm⁻² of Li_xNi_{0.8}Mn_{0.1}Co_{0.1}), and a lean carbonate electrolyte condition (E/C ratio of 4.0 μl mAh⁻¹).

(Line 409-411)

The introduction of NW defect significantly improved the cycling stability of an AF-LMB with high areal capacity (4.2 mAh cm^{-2}) under a lean electrolyte condition (E/C of $4.0 \mu\text{l mAh}^{-1}$) operating 50 cycles with 90 % capacity retention.

11. Line 58: “The defect-free SEI and Li-C orbital hybridization enable lateral Li deposition even in the internal structure of the 3D current collector where no external pressure is exerted normal on its surfaces”

I do not believe there is sufficient evidence to back up this claim. The authors provide top-down SEM images, but nothing that shows the morphology of the lithium in the internal porosity of the current collectors. The cross-sectional images seem to demonstrate that lithium indeed plates within the internal porosity, but it does not evidence that the lithium deposition inside the current collector is lateral. The authors seem to be inferring lateral deposition in the internal porosity from the lateral deposition demonstrated in Fig 5d-f which appears to be on the top surface of the current collector (which would be exposed to external pressure from the coin cell construction). I do not believe the authors have provided sufficient evidence to claim that they have enabled lateral Li deposition in the internal structure of the 3D current collector.

Response: Thanks for the critical comments from the reviewers. We tried to express the behavior of the growth of lithium nuclei on the carbon fiber with the word 'lateral'. This horizontal growth on the surface of the carbon fiber was observed at the initial nucleus growth or small Li deposition capacity about 3.0 mAh cm^{-2} , and then it was observed that the Li grows filling the pore between the fibers, and this is expressed as line 327-330 'By sharp contrast, for d-CP, its carbon fibers were uniformly covered by plated Li at 3 mAh cm^{-2} due to the uniform Li nucleation and lateral growth driven by the NW structure (Fig. 6f). Further deposition led to connection of the Li deposits to those on adjacent fibers (Fig. 6g, h).' We agreed that the expressions at introduction and conclusion section is ambiguous; therefore, it was corrected as 'initial lateral growth of Li nucleus and the resulting dense morphology'.

Revised manuscript:

(Line 6-9)

The nano-window defect-enriched surface induces a defect-free solid-electrolyte interphase by alleviating electron tunneling at the interface, and leads to a uniform Li nucleation and lateral growth of Li nucleus via Li-carbon orbital hybridization, resulting in a dense Li morphology.

(Line 50-53)

the NW structure addresses the critical issues at the anode interface by 1) constructing a thin and uniform SEI with reduced electrolyte decomposition and 2) evenly distributed and highly populated Li nucleus and its lateral growth on the current collector surface.

12. Fig 7d shows the lean electrolyte condition cycling up to 50 cycles. The d-CP show good capacity retention over this range, only 10% loss over 50 cycles (as reported in the abstract). However, Fig S16 shows that there is dramatic capacity roll over by cycle 60. I believe it is disingenuous for the full 100 cycle results to be hidden in the supplementary information. Fig 7c shows the flooded cells up to 100 cycles; the lean electrolyte cells should be shown to 100 cycles in the main paper as well. The challenges of lean electrolyte should not be minimized. The authors should be upfront about this and discuss these challenges in the main text.

Response: We appreciate the reviewer's comment. As the reviewer suggested, Figure 7 was modified. In addition, the current problems of the lean electrolyte cell are described in the main text.

Revised manuscript:

Fig 7: Electrochemical performance of AF-LMB with d-CP under realistic conditions. a, Voltage profiles of the anode-free NCM811 full cell with Cu or d-CP during the charge at 0.1 C and sequential constant voltage charging at 4.3 V (cut-off current: 0.02 C) and discharge. **b,** Comparison of the areal capacity and current density ensuring stable operation for this work and previous works. **c,** The plots of charge and discharge capacity as a function of cycle number for the anode-free NCM811 full cells with 1 M LiPF₆ EC/DEC 10% FEC 1% VC in a flooded cell condition (E/C ratio of 23.8 μl mAh⁻¹) and a lean electrolyte condition (E/C ratio of 4.0 μl mAh⁻¹). The flooded and lean electrolyte cells were cycled at a charge/discharge rate of 0.48 C (2 mA cm⁻²) with a potential window of 2.7-4.3 V.

(Line 364-367)

As displayed in Fig. 7b, the operation current density and the areal capacity of the AF-LMB with d-CP were higher than the previously reported values for liquid electrolyte based AF-LMBs^{2,38-47}, emphasizing the advantage of the NW structure.

(Line 384-385)

The rapid decrease in discharge capacity after 60 cycles is attributed to electrolyte depletion (Supplementary Fig. 20).

13. In their conclusion, the authors state that “a rational design strategy for advancing AF-LMBs is presented.” Can the authors comment about how practical large scale synthesis of these current collectors might be? Without a copper substrate, will these carbon materials be flexible enough to be used in practical format cells (wound pouches or cylinders)? Also, it is well known that lithium electrodes thicken during aging due to the accumulation of dead lithium. With these current collectors be strong enough to accommodate this expansion or might they crumble due to larger internal volume expansion?

Response: We are thankful for the reviewer's advice on improving the quality of this paper. The carbon paper can be bent to some extent, but it is not compatible with winding (bending radius ~5 cm). Therefore, stack-type pouch or prismatic cell are a possible format. For exploiting NW technology for jelly-roll based cells (cylindrical or prismatic cell), the NW defective carbons would be coated on flexible substrates such as polymers or metal foils by a wet coating method. For example, the carbonized ZIF-8 powders and binder are mixed in a solvent and the mixture is coated on a flexible substrate, followed by drying.

Regarding the issue of mechanical deterioration of carbon paper electrode, we agree with the reviewer's concern. In this work, Li deposit ($\leq 5 \text{ mAh cm}^{-2}$) did not completely fill the interstitial pores of d-CP, and thus stress would not be significant. However, at higher deposition capacity can cause the fracture of carbon fiber material. The conclusion "a rational design strategy for advancing AF-LMBs is presented" was made to deliver the efficacy of the concept of low Fermi-level surface. However, we recognized that it can be misinterpreted as the use of carbon fiber is advantageous. Therefore, we revised the conclusion, focusing on the surface design of the anode current collector for advancing AF-LMBs. Also, in the main text, we described the limitations of the carbon paper material used in our work, and the possibility of using the NW technology to modify the surfaces of other advanced carbon material substrates.

Revised manuscript:

(Line 395-398)

From a practical point of view, the use of carbon paper possesses a risk of mechanical deterioration at much larger Li plating capacity or at significant bending, hindering its use for winding-type cells. However, the NW technology could be used to modify the surfaces of various current collectors with higher mechanical strength and flexibility.

(Line 411-414)

Based on the model system provided herein regarding how the electronic structure of the current collector controls the SEI microstructure and Li nucleation, a rational design of surface electronic structure for advancing AF-LMBs is presented.

REVIEWERS' COMMENTS

Reviewer #1 (Remarks to the Author):

I am really impressed with the authors' thorough response to the comments I had raised. The new additions on the theory side make for a really great manuscript. I agree with the authors' assertion that lithium metal batteries should be distinguished from anode-free batteries.

V. Viswanathan

Reviewers' Comments:

Reviewer #1 (Remarks to the Author):

The authors report on an interesting approach to enable anode-free lithium metal batteries. Anode-free lithium metal batteries is the ultimate as it has no excess lithium and provides highest energy density and low cost. However, liquid electrolytes generally react with the current collector and thus, modifying the current collector is a critical challenge being addressed thorough several approaches. Authors use a NW defect-enriched C-based current collector that provides good Li nucleation, growth of Li. The idea is novel, but authors should consider the comments below to improve their work:

- 1) The analysis shown in Fig. 5 is good, but the reference state as Li bulk would be better as that automatically provides a voltage scale. See for example, ACS Energy Lett., 2019, 4, 2952-2959.
- 2) What happens to the surface diffusion barrier for these systems? Do they observe any BEP relationship with the adsorption strength?
- 3) In Fig 7, the authors should compare against a few other data points from PNAS 2020 117 (44) 27195-27203 and the associated data points there.
- 4) Can the authors discuss the rationale for selection of 1 M LiPF₆ ethylene carbonate (EC)/diethyl carbonate (DEC) (1:1 by vol) + FEC (10%) + VC (1%). The electrolyte selection is almost as critical as the current collector, so this deserves some discussion.
- 5) It would be interesting if the authors are able to place all energy levels on the vacuum scale together with where the Li/Li⁺ redox sits in this liquid electrolyte. This could be done through calibration with a redox couple. See for example, J. Phys. Chem. Lett. 2014, 5, 14, 2419–2424.

Reviewer #2 (Remarks to the Author):

The authors report introducing a nano-window defect to carbon paper which lowers the Fermi level to reduce SEI formation due to electron tunneling. This is a thorough study that has many interesting results. However, there are many questions and concerns that must be resolved before publication.

1. Line 31: "While the efficacy of three dimensional (3D) porous current collectors in reducing the effective current density has been proven, their enlarged electroactive surface area can accompany a larger degree of electrolyte decomposition.

The authors highlight the effect of conventional 3D porous current collectors. Can the authors comment on how this effect is contributing to the improved performance of their d-CP current collector and how the benefits obtained from low Fermi-level surface introduced here can be distinguished from the reduction in current density obtained from the porosity of the current collector? The authors demonstrate how p-CP performs better than simple Cu—is that simply the benefit of the current collector porosity, and then then further benefit of the d-CP can be attributed to the NW defects?

The authors also highlight the issue of increased electrolyte decomposition due to the enlarged surface area of porous current collectors. The authors demonstrate how this is not a problem going from p-CP to d-CP which has a higher surface area. However, will this be a problem compared to conventional low surface area current collectors? For example, if the authors wanted to further improve the d-CP performance by using an optimized electrolyte, might this electrolyte degrade faster in a d-CP cell in comparison to a conventional Cu cell?

Furthermore, the authors show that electrolyte dry-out is a major issue for these cells when lean electrolyte conditions are used. Will the increased current collector porosity exacerbate this issue? Can dry-out be avoided with lean electrolytes using these porous current collectors?

2. Line 38: "The mosaic and grain-coalesced SEI formed on the current collector, which originates from aggressive electron tunneling from the current collector to the electrolyte, induces uneven Li nucleation, resulting in protrusion of the Li deposit through the defects in the SEI."..."however, we envisioned that lowering Fermi level at the surface of the current collector can enfeeble the electron tunneling by increasing the tunneling barrier."

Electron tunneling is surely a mechanism for SEI growth. However, it is not clear at all to this reviewer that the SEI formed as a result of electron tunneling is any worse than SEI formed for any other reason, i.e. direct contact with un-passivated lithium due to SEI breaking from large volume expansion and contraction. The entire thesis of this work is founded on the d-CP lowering the Fermi energy for mitigating SEI growth due to electron tunneling. Can the authors explain why avoiding SEI formed via electron tunneling is so important, especially since lithium inventory loss to the reformation of SEI due to large volume expansion and contraction seems more likely for lithium metal cells?

3. The authors should provide the porosity and pore volume of the carbon paper current collectors. The authors qualitatively demonstrate that lithium is plated in the interior volume of the d-CP with top-down and cross-sectional SEM images. It would be useful to know how much lithium could theoretically be accommodated in these current collectors.

4. The authors argue that the benefit of d-CP current collectors is the lowered Fermi level enabled by the NW defects which results in an optimized SEI. Can the authors comment how these NW defects may impact the performance of conventional Li-ion cells? For example, if these NW defects were introduced to conventional Li-ion cell-grade graphites to be used in a conventional Li-ion cell.

5. The authors demonstrate that the first cycle CE of the p-CP and d-CP are 80.8 and 87.4% respectively. These are quite low compared to decent graphites used in Li-ion cells that have first cycle CEs > 90%. Why is this the case? Does this speak to the issue of increased electrochemical surface area?

6. In Fig. 3a, the authors show the first cycle voltage curves of the p-CP and d-CP current collectors. The authors comment that the introduced defects of the d-CP does not significantly impact the delivered capacity. However, these data are plotted vs areal capacity (mAh/cm²). Can the authors also include a specific capacity (mAh/g) scale so readers can easily tell how these capacities compare to normal graphites used in Li-ion cells? If it happens to be lower than the typical ~350 mAh/g, can the authors explain why?

7. In the main text, there is no mention of the loading (mg/cm² and mAh/cm²) of the carbon paper current collectors. It is very important to understand how much capacity is being intercalated into the graphite compared to how much is plated. This information must be included in the main text.

8. The authors do not comment in the main text how the d-CP current collector impacts

volumetric energy density. Replacing an 8 μm Cu current collector with 100 μm d-CP will significantly decrease volumetric energy density. The authors should be forthright with this in the main text.

I also have some issues with the calculated energy densities presented in supplementary 3: The areal capacity used for these calculations is 4.2 mAh/cm^2 , but the capacity demonstrated in Fig 7d is $< 4.0 \text{ mAh/cm}^2$. The average voltage used for all of these calculations is 3.9 V. However, the average voltage for a cell with some graphite will be lower than with a bare copper current collector (this is clearly shown in Fig 7b). The stack used to calculate these numbers only contains a single separator (shown in Fig S17a). However, a bi-cell stack requires two separators (two positive electrodes, two negative electrodes, two separators!). These discrepancies should be corrected.

Furthermore, the authors should include the total stack thickness they use to calculate the volumetric energy density in supplementary table 3. I do not get the same volumetric energy density numbers when I try to calculate it based on the thickness numbers provided in this table. There also seems to be a discrepancy between the provided volumetric energy density of Cu cells vs CP cells. The CP stack thickness is 50% greater than the Cu stack (281 vs 189 μm), but the energy density is only decreased to 80% (1165 vs 1418 Wh/L). If the stack is 1.5x thicker, the volumetric energy density should be 1.5x less.

9. Fig 6a presents coulombic efficiency on a scale from -10 to 110%. This is an unacceptable scale when CEs higher than 95% are required to even approach viability, and differences of just 1% make a very significant difference to lifetime. The minimum CE on this scale should be 90%.

10. Throughout the paper, the authors claim their results have been achieved in an unpressurized system. On line 338 they state:

"It should be stressed that the cycle performance of d-CP in the AF-LMB is not derived from the pressure effect because stack pressure is not delivered to the inner surface of the 3D current collector."

That may be so. However, it is my understanding that these results were achieved in a coin cell format. Unless the coin cells were constructed in some unconventional way which should be reported, then there is significant pressure exerted on the electrodes which will improve Li metal reversibility. Even if the authors' above argument is true, there will be pressure exerted on the outer surfaces of the porous current collectors and Li plating there. Regardless, if the authors want to make the claim of achieving results in an unpressurized cell, they must present results that were obtained in a truly unpressurized cell (an unconstrained pouch, for example). The authors need to explain if these results are indeed from a conventional coin cell that has increased uniaxial pressure, and if so, remove or highly temper the claims of "unpressurized" results.

11. Line 58: "The defect-free SEI and Li-C orbital hybridization enable lateral Li deposition even in the internal structure of the 3D current collector where no external pressure is exerted normal on its surfaces"

I do not believe there is sufficient evidence to back up this claim. The authors provide top-down SEM images, but nothing that shows the morphology of the lithium in the internal porosity of the current collectors. The cross-sectional images seem to demonstrate that lithium indeed plates within the internal porosity, but it does not provide evidence that the lithium deposition inside the current collector is lateral. The authors seem to be inferring lateral

deposition in the internal porosity from the lateral deposition demonstrated in Fig 5d-f which appears to be on the top surface of the current collector (which would be exposed to external pressure from the coin cell construction). I do not believe the authors have provided sufficient evidence to claim that they have enabled lateral Li deposition in the internal structure of the 3D current collector.

12. Fig 7d shows the lean electrolyte condition cycling up to 50 cycles. The d-CP show good capacity retention over this range, only 10% loss over 50 cycles (as reported in the abstract). However, Fig S16 shows that there is dramatic capacity roll over by cycle 60. I believe it is disingenuous for the full 100 cycle results to be hidden in the supplementary information. Fig 7c shows the flooded cells up to 100 cycles; the lean electrolyte cells should be shown to 100 cycles in the main paper as well. The challenges of lean electrolyte should not be minimized. The authors should be upfront about this and discuss these challenges in the main text.

13. In their conclusion, the authors state that "a rational design strategy for advancing AF-LMBs is presented." Can the authors comment about how practical large scale synthesis of these current collectors might be? Without a copper substrate, will these carbon materials be flexible enough to be used in practical format cells (wound pouches or cylinders)? Also, it is well known that lithium electrodes thicken during aging due to the accumulation of dead lithium. With these current collectors be strong enough to accommodate this expansion or might they crumble due to larger internal volume expansion?

Response Letter

We thank for peer reviewing our manuscript and giving helpful and critical comments and advices. The constructive comments from the reviewer have significantly improved the quality of this manuscript. We have done further investigations about binding energies, diffusional behaviors of Li adatom on various surface and representation of Fermi level in main figures reflecting reviewers' advices. In addition, through deep contemplation of the reviewer's questions, we tried to clarify 1) the merits of a three-dimensional current collector with a low Fermi level that it has vaguely described in our previous manuscript and 2) that this current collector can have in a practical viewpoint. We hope that this revision can fulfill the reviewers' comments. The changes made are indicated by **yellow background** in the revised manuscript.

Title: Driving lean-electrolyte anode-free Li metal batteries by an electron-deficient three-dimensional current collector

Reviewer #1

1. The analysis shown in Fig. 5 is good, but the **reference state as Li bulk would be better as that automatically provides a voltage scale**. See for example, ACS Energy Lett., 2019, 4, 2952-2959.

Response: We appreciate for suggesting the paper that is closely related to our research. The binding energy shown in Figure.5I was obtained by subtracting the energy of each of the substrate and Li adatom from the energy of the structure in which Li adatom is adsorbed on the substrate, as written in Supplementary Note 1. All the calculated energies correspond to the total energy of each structure at 0 K, which is the same as gibbs free energy or enthalpy assuming that temperature contribution to free energy is negligible (ACS Energy Lett., 2019, 4, 2952-2959). Therefore, they are equivalent to the formula used to calculate nucleation overpotential presented in the suggested reference. In order to convey a clearer meaning, the term 'binding energy' was replaced with 'adsorption free energy, ΔG_{ads} ' in the revised manuscript, and the suggested reference (ref. 33 in the manuscript) was added to the revised manuscript.

2. What happens to the surface **diffusion barrier for these systems?** Do they observe any BEP relationship with the adsorption strength?

Fig. R1: Diffusional behavior of Li adatom on various substrates. a-d, Calculated Li adatom surface diffusion activation energies on (a) Li(111), (b) Li(110), (c) Li(200), (d) BLG. The blue atoms denote diffusing Li adatom.

e, Energy barrier for diffusion of Li adatom from inside to outside of NW-Li structure. **f**, Brønsted–Evans–Polanyi (BEP) relation between calculated adsorption enthalpy (ΔH_{abs}) and diffusion activation energy (E_{diff}) of Li adatom on various substrate.

Response: We are grateful for what the reviewer’s comment. As suggested, we calculated the diffusion barrier of lithium adatom on various surface structures. In the case of lithium's dominant crystal plane ((200), (110), (111)) and BLG, the diffusion barrier was calculated through transition state (TS) search. In the case of the Li-NW structure, it was difficult to define a diffusion path, so the energy difference of inside and outside of defect (Fig. 51) was considered as activation barrier for diffusing from inside of defect to outside. As the reviewer commented, our calculation results showed a linear relationship between adsorption enthalpy and diffusion barrier.

Through BEP relationship analysis, it was clarified that the current collector with strong binding energy can suppress the surface diffusion of lithium adatom on the surface. Owing to the result, we are able to more clearly deliver our claim; contrary to the rapid surface Li diffusion and consequent formation of non-uniform Li cluster on the graphene surface, the NW-Li structure does not bring coalesce of Li nuclei due to the limited surface Li diffusion.

These new data would be meaningful to potential readers of our manuscript, thus were added the data and reference to the revised manuscript.

Revised manuscript:

(Supplementary Fig. 15)

Supplementary Figure 15. Diffusional behavior of Li adatom on various substrates. a-e, Calculated Li adatom surface diffusion activation energies on (a) Li(111), (b) Li(110), (c) Li(200), (d) BLG, and (e) NW-Li. The blue atoms denote diffusing Li adatom. **f**, Brønsted–Evans–Polanyi (BEP) relation¹⁴ between calculated adsorption enthalpy and diffusion activation energy of Li adatom on various substrate.

3. In Fig 7, the authors should compare against a few other data points from PNAS 2020 117 (44) 27195-27203 and the associated data points there.

Response: The performance comparison in Fig. 7 was intended to make fair comparison among AF-LMBs. The suggested reference deals with lithium metal battery. Thus, it may be not appropriate to include the suggested data points in Fig. 7. We hope the reviewer understand this point. The suggested reference is a highly valuable reference for delivering the current state of art. In this aspect, it was referenced in the introduction section of the revised manuscript (ref. 15 in the manuscript).

4. Can the authors discuss the rationale for selection of 1 M LiPF₆ ethylene carbonate (EC)/diethyl carbonate (DEC) (1:1 by vol) + FEC (10%) + VC (1%). The electrolyte selection is almost as critical as the current collector, so this deserves some discussion.

Response: We highly appreciated the reviewer's comments for our manuscript and strongly agree that it is worth discussing about the rationality of electrolyte selection. In our anode-free battery, since lithiation/de-lithiation of graphite as well as plating/stripping of lithium occurs, carbonate solvents (EC, DEC and FEC) were selected to prevent exfoliation of graphite rather than etheral solvents (Adv. Energy Mater. 2017, 7, 1700418). For the other reason, we targeted to employ high capacity NCM electrode; the high voltage cathode deters etheral electrolytes but prefers carbonate electrolytes. The LiPF₆ salt was chosen to only focus on the performance change depending on the current collector when using the most commonly used lithium salt. 2% of VC was added to stabilize initial surface film (SEI and CEI) structure on anode and cathode (J. Electrochem. Soc. 2009, 156, A103-A113). We added the discussion to the revised manuscript.

Revised manuscript:

(Line 357-362)

In our anode-free battery, since lithiation/de-lithiation of graphite as well as plating/stripping of lithium occurs, carbonate solvents (EC, DEC and FEC) were selected to prevent exfoliation of graphite rather than etheral solvents³⁶. In addition, the carbonate electrolytes enable the use of NCM cathode due to their high oxidation stability. LiPF₆ was chosen due to its high ionic conductivity in carbonate solvents. 2% of VC was added to help the formation of stable SEI and cathode-electrolyte interphase (CEI)³⁷.

5. It would be interesting if the authors are able to place all energy levels on the vacuum scale together with where the Li/Li⁺ redox sits in this liquid electrolyte. This could be done through calibration with a redox couple. See for example, J. Phys. Chem. Lett. 2014, 5, 14, 2419–2424.

Fig. R2: Experimental and simulated Fermi-level landscape during lithiation. **a**, Plots of Fermi level for p-CP and d-CP during lithiation from OCV to 1.0 V (V vs. Li/Li⁺) determined from extrapolation of the secondary cut-off region in UPS. All the Fermi levels of Li/Li⁺, redox associated with electrolyte reduction, Li metal, p-CP and d-CP were calibrated with vacuum state of E_{vac}=0 eV. **b**, Calculated surface Fermi-level and atomic structure of BLG (green plots), NW-Li (red plots) and calculated redox potential of lithiation of NW (blue plots).

Response: We thank for the reviewer's detailed comments to improve the quality of our manuscript. Since electrode Fermi level is a critical factor to the redox reactions of Li/Li⁺ and electrolyte decomposition, displaying redox potentials and Fermi level of electrode surface on vacuum level scale altogether may increase reader's understanding. As the reviewer suggested, we revised the figure and added the provided reference (ref. 27 in the manuscript).

In addition, to support the UPS result, we newly calculated Fermi levels of lithiated BLGs at different degree of lithiation and added them to new figure. In line with the UPS result, the NW structure has a lower Fermi level than BLG at all the lithiation degrees. Based on the additional results, we revised Fig. 4 and related main text.

Revised manuscript:

(Figure 4)

Fig. 4: Atomistic and electronic structure of NW defect during lithiation. **a**, Voltage profiles for the early stage of galvanostatic lithiation for p-CP and d-CP at 0.1 mA cm⁻² in half-cell configuration. **b, c**, Image plots of operando XRD spectra along with the potential profile during the first lithiation and deconvoluted XRD patterns at OCV, 1.5 V and 0.8 V for p-CP (b) and d-CP (c). The peaks from the lithiated defect structure (lithiated NW, Li₃C₈) at 43° of 2θ are indicated by blue lines. **d**, Plots of Fermi

level for p-CP and d-CP during lithiation from OCV to 1.0 V (V vs. Li/Li⁺) determined from extrapolation of the secondary cut-off region in UPS. All the Fermi levels of Li/Li⁺, redox associated with electrolyte reduction, Li metal, p-CP and d-CP were calibrated with vacuum state of $E_{\text{vac}}=0$ eV. e, The relaxed structures of lithiated BLG, NW and fully lithiated NW (Li₃C₈), corresponding DFT calculations of Fermi level and redox potential for NW at various lithiation degrees. The Li, carbon at the upper layer, and carbon at the lower layer are shown in blue, yellow, and gray color, respectively.

(Line 177-213)

The atomic structural changes of p-CP and d-CP during the early stage lithiation were traced by an operando X-ray diffraction (XRD) analysis (Fig. 4b, c). The XRD image plots in the 24-30° region shed light on the lithiation process of the graphitic structure of carbon fiber. In contrast to the strong peak at 2θ of 26.5° from G(002) of p-CP, the weak intensities at around 26.5° for d-CP indicate the existence of a defective carbon layer. For d-CP, the lithiated structures of LiC₆ and LiC₁₂ started to appear at lower lithiation capacities than for p-CP, suggesting that relatively fast Li transport through the defective layer facilitates the lithiation of the inner graphitic carbon fiber. The structural evolution of the defect structure can be traced from the higher 2θ region. From the beginning of the lithiation, a peak at $2\theta=43^\circ$ appeared for d-CP and gradually grew with decreasing potential, indicating Li insertion into the NW defect structure of the graphite surface^{24,25}. By contrast, the XRD pattern of p-CP was invariant in the potential range where solid-solution type lithiation at the 1L stage of initial graphite lithiation (Li_xC₆, $x<0.16$)²⁶ occurs; the solid-solution type lithiation does not change the lattice structure of graphite surface.

In order to trace the electronic structure along with the variation of surface atomic structure, the work function, which is the difference between the vacuum level (E_{vac}) and Fermi level²⁷, was measured along the potential descending for p-CP and d-CP by using UV photoelectron spectroscopy (UPS, Supplementary Fig. 4). The work function of the intrinsic electrode surface of p-CP is 4.52 eV, which is coincident with the work function of the graphite reported in previous studies²⁸, and that of d-CP is 4.80 eV (Fig. 4d). Fermi levels of p-CP and d-CP increased with decreasing potential upon lithiation. Notably, Fermi levels of d-CP were much lower than those of p-CP regardless of the degree of lithiation, indicating that the surface of d-CP has a lower electron energy level upon charging.

In line with the UPS result, DFT calculation also demonstrated that the NW structure has a lower Fermi level than BLG during lithiation. Based on the 2D/G ratio in raman spectroscopy (Fig. 2h) and pore size distribution (Fig. 2i), we constructed bilayer graphene (BLG) and NW defect in BLG (Supplementary Fig. 5, 6). According to the calculations of the sequential lithiation of NW, a Li atom spontaneously penetrates through the NW hole, accepting electrons of Li by the electron-deficient NW (Supplementary Fig. 7, 8) and places between the graphene layers with a gradual diminution of spin momentum (Supplementary Fig. 9, 10) until a stoichiometry of Li₃C₈ (fully lithiated state) is reached. The calculated potentials profile and Fermi levels of Li-NW structures at different degrees of lithiation are shown in Fig. 4e. The potential change with lithiation accounts for the larger lithiation capacity of d-CP above 1.5 V vs. Li/Li⁺. In accordance with the UPS results, the calculated Fermi levels of the NW structure were much lower than those of BLG. The results indicate that the Fermi level of d-CP is lower than that of p-CP even though the NW was lithiated, and the electron-deficient properties of d-CP caused a change in the surface SEI structure.

Reviewer #2

1. Line 31: “While the efficacy of three dimensional (3D) porous current collectors in reducing the effective current density has been proven, their enlarged electroactive surface area can accompany a larger degree of electrolyte decomposition.

The authors highlight the effect of conventional 3D porous current collectors. Can the authors comment on how this effect is contributing to the improved performance of their d-CP current collector and how the benefits obtained from low Fermi-level surface introduced here can be distinguished from the reduction in current density obtained from the porosity of the current collector? The authors demonstrate how p-CP performs better than simple Cu—is that simply the benefit of the current collector porosity, and then then further benefit of the d-CP can be attributed to the NW defects?

The authors also highlight the issue of increased electrolyte decomposition due to the enlarged surface area of porous current collectors. The authors demonstrate how this is not a problem going from p-CP to d-CP which has a higher surface area. However, will this be a problem compared to conventional low surface area current collectors? For example, if the authors wanted to further improve the d-CP performance by using an optimized electrolyte, might this electrolyte degrade faster in a d-CP cell in comparison to a conventional Cu cell?

Furthermore, the authors show that electrolyte dry-out is a major issue for these cells when lean electrolyte conditions are used. Will the increased current collector porosity exacerbate this issue? Can dry-out be avoided with lean electrolytes using these porous current collectors?

Fig. R3: **a**, EIS analysis for the half-cells with copper, p-CP and d-CP after the pre-cycle. **b, c**, Quantification of irreversible capacity caused by Li metal plating/stripping (**b**) or sequential restoration of damaged SEI on current collector surface (**c**). The restoration of SEI was performed by galvanostatic cycling at a current density of 0.1 mA cm⁻² in the range of 0 to 2.0 V.

Fig. R4: **a-c**, SEM images after 10 cycles of 5 mAh cm⁻² Li deposition at 2 mA cm⁻² for Cu (**a**), p-CP (**b**) and d-CP (**c**). Electrolyte: 1 M LiPF₆ EC/DEC 1:1 v/v 10% FEC 1% VC

Response:

The authors highlight the effect of conventional 3D porous current collectors. Can the authors comment on how this effect is contributing to the improved performance of their d-CP current collector and how the benefits obtained from low Fermi-level surface introduced here can be distinguished from the reduction in current density obtained from the porosity of the current collector? The authors demonstrate how p-CP performs better than simple Cu—is that simply the benefit of the current collector porosity, and then then further benefit of the d-CP can be attributed to the NW defects?

We appreciate the reviewer's constructive question. In the previous our manuscript as the reviewer said, we simply explain that the reason why p-CP shows better performance than Cu is because of its three-dimensional structure. Complete decoupling of the current density effect and low-Fermi level effect is highly challenging. However, we can assure that the good performance of d-CP is not due to the current density effect. It is demonstrated by the comparison of impedance and irreversible capacity loss between Cu and p-CP. As shown in Fig. R3a, the interfacial resistance of p-CP is much smaller than that of Cu, indicating that effective surface area is a dominating factor for interfacial resistance as reported in a previous literature (*Adv. Funct. Mater.* 2017, 27, 1606422). However, the coulombic efficiency for Li deposition and dissolution was quite similar between p-CP (89.2%) and Cu (90.2%) as shown in Fig. R3b, which imply that the irreversible capacity due to the large surface area offset the current density effect. It is supported by the subsequent cycle in the range of 0-2.0 V. The irreversible capacity found in the subsequent cycle comes from the restoration of SEI by electrolyte decomposition. The Cu and p-CP showed 0 and 0.143 mAh cm⁻² of irreversible capacity, verifying intensive electrolyte decomposition on 3D current collector. Therefore, the benefit of the current collector porosity cannot be the reason for the outstanding performance of d-CP.

The authors also highlight the issue of increased electrolyte decomposition due to the enlarged surface area of porous current collectors. The authors demonstrate how this is not a problem going from p-CP to d-CP which has a higher surface area. However, will this be a problem compared to conventional low surface area current collectors? For example, if the authors wanted to further improve the d-CP performance by using an optimized electrolyte, might this electrolyte degrade faster in a d-CP cell in comparison to a conventional Cu cell?

We appreciate the reviewer for the valuable questions. Because the anode-free electrode is a complex system, including electrolyte decomposition effect at not only current collector but also Li deposit and geometry effect, it is hard for us to make a concrete answer to the reviewer's question. However, we would like to deliver your insights on the interplay among the various effects in the following.

As the reviewer commented, the electrolyte decomposition at low surface area current collector surface is not as serious as 3D current collector. However, as observed for Cu current collector, its reversibility of Li plating/stripping is poor due to the inhomogeneous Li plating. It means that, for conventional low surface area current collector, the issue of Li deposition morphology and consequent electrolyte decomposition at Li deposit would be more important than the amount of electrolyte decomposition at Cu surface.

Although the irreversible capacity (Li plating/stripping) for d-CP is much smaller than that for Cu (Fig. R3b), there are more electrolyte decomposition on d-CP surface due to the larger surface area (Fig. R3c). If the electrolyte decomposition at current collector surface can be mitigated by using an optimized electrolyte as exemplified by the reviewer, the advantage of 3D current collector (reduction of effective current density) can be also pronounced, possibly resulting in a better performance.

Recent electrolyte optimization is directed to form a stable SEI by means of electrolyte decomposition (*Nat. Energy* 2021, doi: 10.1038/s41560-021-00787-9). Therefore, gradual electrolyte consumption is inevitable. However, in the case of d-CP, Li morphology can be controlled by surface electronic property with minimizing the electrolyte consumption. In this aspect, we expect that electrolyte optimization and low-Fermi level surface can be complementary; the optimized electrolyte forms a stable SEI and low-Fermi level surface minimize its expense.

Furthermore, the authors show that electrolyte dry-out is a major issue for these cells when lean electrolyte conditions are used. Will the increased current collector porosity exacerbate this issue? Can dry-out be avoided with lean electrolytes using these porous current collectors?

The electrolyte dry out is influenced not only the geometry of current collector but also roughness of Li deposit and dead-Li layer. Even though current collector porosity exacerbates this issue, it can be offset or

overcompensated by minimizing electrolyte decomposition on current collector and inducing uniform Li deposit as demonstrated with d-CP in this work. In contrast to d-CP, Cu and p-CP form a thick dead-Li layer after cycling (Fig. R4), aggregating the issue of electrolyte dry out.

After contemplating the reviewer's comments, we added the data corresponding to Cu (Fig. R3b and c) was added to Fig. 6j and k to describe the comparison with the 2D current collector in more detail.

Revised manuscript:

(Figure 6)

Fig 6: Reversibility of Li plating/stripping on the defective carbon surface. **a**, The plots of CE as a function of cycle number for Cu, p-CP, and d-CP current collector in half-cell configuration at a current density of 2.0 mA cm⁻² and a capacity of 5.0 mAh cm⁻². **b-i**, Li deposition morphologies for p-CP (b-e) and d-CP (f-i) at various plating capacities from 3.0 to 9.0 mAh cm⁻². **j, k**, Quantification of the irreversible capacity caused by Li metal plating/stripping and sequential restoration of damaged SEI on current collector surface. The restoration of SEI was performed by a galvanostatic cycling in the range of 0.0 and 2.0 V at a current density of 0.1 mA cm⁻². **l, m**, Schematics for the irreversible loss of Li and the electrolyte on p-CP surface (l), and the proposed mechanism for prevention of the irreversibility on d-CP surface (m).

(Line 331-337)

The irreversible capacity of the first cycling originates from the formation of SEI on current collector and deposited Li and of isolated Li, whereas that of the second partial cycling should reflect the restoration of the SEI on the current collector damaged during the first cycle. The capacity loss from the Li plating/stripping was 0.490 mAh cm⁻² (CE of 90.2 %) for Cu and 0.541 mAh cm⁻² (CE of 89.2 %) for p-CP (Fig. 6j) which points out that larger amount of Li was lost with p-CP and that the SEI formation on its large surface area offsets the positive effect of a reduced current density.

2. Line 38: “The mosaic and grain-coalesced SEI formed on the current collector, which originates from aggressive electron tunneling from the current collector to the electrolyte, induces uneven Li nucleation, resulting in protrusion of the Li deposit through the defects in the SEI.”...”however, we envisioned that lowering Fermi level at the surface of the current collector can enfeeble the electron tunneling by increasing the tunneling barrier.”

Electron tunneling is surely a mechanism for SEI growth. However, it is not clear at all to this reviewer that the SEI formed as a result of electron tunneling is any worse than SEI formed for any other reason, i.e. direct contact with un-passivated lithium due to SEI breaking from large volume expansion and contraction. The entire thesis of this work is founded on the d-CP lowering the Fermi energy for mitigating SEI growth due to electron tunneling. Can the authors explain why avoiding SEI formed via electron tunneling is so important, especially since lithium inventory loss to the reformation of SEI due to large volume expansion and contraction seems more likely for lithium metal cells?

Response: We appreciated the reviewer for critical comments for helping us improve logic of this manuscript. In anode-free lithium batteries, we agree with the reviewer's comment that dead-Li and a continuous reaction of lithium and electrolyte are directly connected to capacity deterioration. As the reviewer commented, according to Figure 6j and k, the irreversible capacity caused by dead-Li and the side reaction is larger than that formed by electron tunneling on the current collector before Li deposition. However, we would like to emphasize that the uniformity of SEI layer on the current collector formed by electron tunneling from the current collector to the electrolyte not only influences the SEI breakage and reformation, but also subsequent uniform initial Li nucleation and growth. The uniform flux of Li⁺ thorough the uniform SEI on current collector is a prerequisite for uniform Li nucleation. As the uniformity of Li deposit is increased, the Li inventory loss is decreased due to a reduced interfacial stress. We agree that this part was ambiguous in the previous manuscript, thus revised to be clearer.

Revised manuscript:

(Line 35-41)

The properties of the mosaic and grain-coalesced SEI formed on the current collector surface, which originate from aggressive electron tunneling from the current collector to the electrolyte, can induces continuous electrolyte decomposition and defective SEI and thereby uneven initial Li nucleation and growth. In this regard, we envisioned that it is possible to modify the microstructure of SEI and the behavior of Li nucleation/growth via tuning Fermi level of the current collector surface.

3. The authors should provide the porosity and pore volume of the carbon paper current collectors. The authors qualitatively demonstrate that lithium is plated in the interior volume of the d-CP with top-down and cross-sectional SEM images. It would be useful to know how much lithium could theoretically be accommodated in these current collectors.

Electrodes	Apparent density (g cm ⁻³)	Carbon fiber density (g cm ⁻³)	Porosity	Pore volume (cm ³ per 1 cm ²)	Maximum Li accommodation (mAh cm ⁻²)
p-CP	0.55	2.26	0.757	7.57x10 ⁻³	15.6
d-CP	0.39	2.26	0.827	8.27x10 ⁻³	17.1

Table R1: Calculated electrode porosity and the maximum capacity of lithium metal that can be stored in the pores of the electrode.

Fig. R5: a, b, SEM images of p-CP (a), d-CP (b) after Li deposition at 0.5 mA cm^{-2} and 16 mAh cm^{-2}

Response: We thank the reviewer for the comments. We calculated the porosity, pore volume, and maximum Li accommodation for p-CP and d-CP from the apparent density of p-CP and d-CP, carbon fiber density, and Li metal density, and listed them in Table R1. We added the data to revised manuscript and supplementary information.

Revised manuscript:

(Line 326-328)

Unlike p-CP, top plating was not observed for d-CP even at the Li deposition capacity (16 mAh cm^{-2}) which corresponds to full accommodation of its pores (Supplementary Table 3 and Supplementary Fig. 18).

(Supplementary Information)

Electrodes	Apparent density (g cm^{-3})	Carbon fiber density (g cm^{-3})	Porosity	Pore volume ($\text{cm}^3 \text{ per } 1 \text{ cm}^2$)	Maximum Li accommodation (mAh cm^{-2})
p-CP	0.55	2.26	0.757-	7.57×10^{-3}	15.6
d-CP	0.39	2.26	0.827	8.27×10^{-3}	17.1

Supplementary Table 3. Calculated electrode porosity and the maximum capacity of lithium metal storage in the pores of the electrodes.

Supplementary Figure 18. a, b) SEM images of a) p-CP , b) d-CP after Li deposition at 0.5 mA cm^{-2} and 16 mAh cm^{-2}

4. The authors argue that the benefit of d-CP current collectors is the lowered Fermi level enabled by the NW defects which results in an optimized SEI. Can the authors comment how these NW defects may impact the performance of conventional Li-ion cells? For example, if these NW defects were introduced to conventional Li-ion cell-grade graphites to be used in a conventional Li-ion cell.

Fig. R6: Coulombic efficiency test of carbon current collector||Li half cell

Response: We appreciate the reviewers' wide view. To check the possibility of improving efficiency for intercalation chemistry with NW, we investigated the reversibility of Li||p-CP and Li||d-CP half-cell with a 1 M LiPF₆ EC/DEC (1:1 v/v). As shown in Fig. R6, the intercalation/de-intercalation efficiency of d-CP was nearly 100%, while p-CP electrode showed a lower CE of 99.68%. The result suggests that the uniform SEI structure derived by NW defects also works for Li-ion batteries.

Revised manuscript:

(Line 136-137)

As a graphite electrode, d-CP showed higher CEs than p-CP for Li intercalation/de-intercalation process, verifying the lowered electrolyte decomposition (Supplementary Fig. 3).

(Supplementary information)

Supplementary Figure 3. Coulombic efficiency test of carbon current collector||Li half cell

5. The authors demonstrate that the first cycle CE of the p-CP and d-CP are 80.8 and 87.4% respectively. These are quite low compared to decent graphities used in Li-ion cells that have first cycle CEs > 90%. Why is this the case? Does this speak to the issue of increased electrochemical surface area?

Response: We are grateful for what the reviewer comments. It is true that carbon paper (p-CP) used in our research has a relatively lower ICE value than the of a graphite electrode (~85%, Nano energy 2020, 77, 105143). However, when calculating the irreversible specific capacity, p-CP is about 43.6 mAh g⁻¹ and d-

CP is 40.4 mAh g^{-1} , which is similar to the first cycle irreversible capacity reported in other papers ($40\text{--}100 \text{ mAh g}^{-1}$, Solid State Ionics 2005, 176, 905-909). In the case of carbon paper, since the reversible capacity is smaller than that of typical graphite (details described in question 6), the ICE value can be lower in spite of similar irreversible capacity.

6. In Fig. 3a, the authors show the first cycle voltage curves of the p-CP and d-CP current collectors. The authors comment that the introduced defects of the d-CP does not significantly impact the delivered capacity. However, these data are plotted vs areal capacity (mAh/cm^2). Can the authors also include a specific capacity (mAh/g) scale so readers can easily tell how these capacities compare to normal graphites used in Li-ion cells? If it happens to be lower than the typical $\sim 350 \text{ mAh/g}$, can the authors explain why?

Fig. R7: Specific capacity of Li intercalation into carbon paper. a, p-CP b, d-CP.

Response: We appreciate the comments and agree with the reviewer's proposal. We measured the specific capacity according to the current density (0.1 mA cm^{-2} , 0.5 mA cm^{-2} , 2.0 mA cm^{-2}) because the intercalation capacity of carbon paper differs with current density. Carbon paper exposes basal plane more than edge plane, so its specific capacity can be smaller than 300 mAh g^{-1} (Nat. Commun. 2017, 8, 812). The charge/discharge voltage profile for p-CP and d-CP was shown in Fig. R7a and b, respectively. The measured specific capacity was 165, 119 and 55.2 mAh g^{-1} for p-CP and 245, 164 and 83.4 mAh g^{-1} for d-CP at 0.1, 0.5 and 2.0 mA cm^{-2} , respectively. d-CP has higher specific capacities due to a promoted Li insertion through the defect hole (Line 193-195). These data were added to the supplementary information of the revised manuscript.

Revised manuscript:

Supplementary Figure 19. Li intercalation and deintercalation potential profiles at various current densities for a, p-CP b, d-CP

According to the half-cell tests (Supplementary Fig. 19) and areal loading (Supplementary Table 3), the areal intercalation capacity is 0.908, 0.654 and 0.304 mAh cm⁻² for p-CP and 0.956, 0.651 and 0.322 mAh cm⁻² for d-CP at 0.1, 0.5, and 2.0 mA cm⁻², respectively. The Li plating capacity for 2.0 mA cm⁻² and 4 mA cm⁻² in anode-free cell operation is 3.70 mAh cm⁻² for p-CP and 3.68 mAh cm⁻² for d-CP.

7. In the main text, there is no mention of the loading (mg/cm² and mAh/cm²) of the carbon paper current collectors. It is very important to understand how much capacity is being intercalated into the graphite compared to how much is plated. This information must be included in the main text.

Response: We appreciate the reviewer's detailed comment on intercalation capacity and deposition capacity. According to the half cell tests (Fig. R7), the areal intercalation capacity is 0.908, 0.654 and 0.304 mAh cm⁻² for p-CP and 0.956, 0.651 and 0.322 mAh cm⁻² for d-CP at 0.1, 0.5, and 2.0 mA cm⁻², respectively. The Li plating capacity for 2.0 mA cm⁻² and 4 mA cm⁻² in anode-free cell operation is 3.70 mAh cm⁻² for p-CP and 3.68 mAh cm⁻² for d-CP. Because areal intercalation capacity (mAh cm⁻²) is important when using carbon as an anode current collector, as suggested by the reviewer, the areal capacity of p-CP and d-CP were included in the main text with a proper supplementary figure.

Revised manuscript:

(Line 368-371)

Figure 7c shows the cycling performance of the anode-free cells at practical charging and discharging current density of 2 mA cm⁻² (0.48 C) under the flooded and lean electrolyte condition. The Li plating capacity and intercalation capacity for the operation were 3.70 and 0.30 mAh cm⁻² for p-CP, and 3.68 and 0.32 mAh cm⁻² for d-CP (Supplementary Fig. 19).

8. The authors do not comment in the main text how the d-CP current collector impacts volumetric energy density. Replacing an 8 um Cu current collector with 100 um d-CP will significantly decrease volumetric energy density. The authors should be forthright with this in the main text.

I also have some issues with the calculated energy densities presented in supplementary 3: The areal capacity used for these calculations is 4.2 mAh/cm², but the capacity demonstrated in Fig 7d is < 4.0 mAh/cm². The average voltage used for all of these calculations is 3.9 V. However, the average voltage for a cell with some graphite will be lower than with a bare copper current collector (this is clearly shown in Fig 7b). The stack used to calculate these numbers only contains a single separator (shown in Fig S17a). However, a bi-cell stack requires two separators (two positive electrodes, two negative electrodes, two separators!). These discrepancies should be corrected.

Furthermore, the authors should include the total stack thickness they use to calculate the volumetric energy density in supplementary table 3. I do not get the same volumetric energy density numbers when I try to calculate it based on the thickness numbers provided in this table. There also seems to be a discrepancy between the provided volumetric energy density of Cu cells vs CP cells. The CP stack thickness is 50% greater than the Cu stack (281 vs 189 um), but the energy density is only decreased to 80% (1165 vs 1418 Wh/L). If the stack is 1.5x thicker, the volumetric energy density should be 1.5x less.

Response: We are very grateful for the reviewer's critical comment and reviewing energy density calculation described in our manuscript. As the reviewer commented, the volumetric energy density of discharged anode-free cell is higher for copper current collector than carbon current collector. However, for Cu current collector, the volumetric energy density of charged state is lower than that of discharged state, which is contrasted by indifference for 3D current collector. Furthermore, ramified Li deposition and build up of porous layer on Cu current collector diminish the volumetric energy density of charged cell upon cycling. Therefore, for fair comparison, the volumetric energy density was calculated using the

thickness of negative electrode at fully charged state as reported in previous literature (Nat. Energy 2020, 5, 693-702).

Assuming dendrite-free Li deposition on Cu current collector, the anode thickness at fully charged state (4.2 mAh cm⁻² at 0.1 C, and 5 μm per 1 mAh cm⁻²) was 50 μm (8μm (Cu) +21μm (Li)*2 (bi cell)) for the copper anode. In the case of p-CP and d-CP, the anode thickness was calculated as 100 μm assuming that 4.2 mAh cm⁻² of lithium was electrodeposited inside the carbon structure. In the case of d-CP, this is true as shown in Supplementary Fig. 14l, but in the case of p-CP, as shown in Supplementary Fig. 14k, a slight thickening of the negative electrode actually occurred during the charging process. The calculated volumetric densities were stated in the main text of the revised manuscript.

Following the indication by reviewer, discharge capacity and average voltage were re-measured at 0.1 C thoroughly; the average voltage of Cu, p-CP and d-CP were 3.860, 3.844, and 3.850 V. The number of PP separator for calculating the energy densities presented in Supplementary Table 3 was correct as two, but the scheme of Supplementary Fig. 17a and relevant description were incorrect. We added a 0.1 C charge/discharge profile (Fig. 7a, revised figure is provided in below question 12), description on how energy density is calculated, the modified the scheme (Supplementary Fig. 21a), and corrected supplementary table 4 in the revised manuscript.

Revised manuscript:

(Line 362-364)

The voltage profiles of the anode-free cell with p-CP or d-CP (Fig. 7a) showed a distinctive carbon lithiation step at the beginning of charging (shaded in blue), as observed for the half cells.

(Line 386-395)

The cell-level energy density of AF-LMBs with different current collector was estimated based on the voltage profiles shown in Fig. 7a. The AF-LMB with d-CP can deliver energy densities of 349.1 Wh kg⁻¹ and 1084.7 Wh l⁻¹. Assuming void-free Li deposition (5 μm per 1.0 mAh cm⁻² Li), the cell with the Cu current collector has a lower gravimetric energy density (338.2 Wh kg⁻¹) but a higher volumetric energy density (1304 Wh l⁻¹) compared with the d-CP cell (Supplementary Table 4, Supplementary Fig. 21). However, when the increased electrode thickness due to the Li dendritic growth and the formation of a thick porous layer is reflected in the calculation, the volumetric energy density of the Cu cell after 10 cycle is lowered to 790.8 Wh l⁻¹, which is contrasted by the invariant energy density for d-CP (Supplementary Fig. 22).

(Supplementary Information)

	Cu	p-CP	d-CP
Al current collector (13 μm)	3.51 mg cm ⁻²	3.51 mg cm ⁻²	3.51 mg cm ⁻²
Anode current collector	7.17 mg cm ⁻² (8 μm)	5.5 mg cm ⁻² (100 μm)	3.9 mg cm ⁻² (100 μm)
NMC811 active material (74 μm)	20.79 mg cm ⁻²	20.79 mg cm ⁻²	20.79 mg cm ⁻²
PP separator (20 μm)	1.95 mg cm ⁻²	1.95 mg cm ⁻²	1.95 mg cm ⁻²
Electrolyte	E/C=4.84 g Ah ⁻¹	E/C=4.84 g Ah ⁻¹	E/C=4.84 g Ah ⁻¹
0.1 C discharge capacity	4.24 mAh cm ⁻²	4.14 mAh cm ⁻²	4.24 mAh cm ⁻²
Average voltage	3.860 V	3.844 V	3.850 V
Total stack energy	32.74 mWh cm ⁻²	31.83 mWh cm ⁻²	32.65 mWh cm ⁻²
Total stack weight	96.8 mg cm ⁻²	93.5 mg cm ⁻²	93.5 mg cm ⁻²
Total stack thickness (Charged)	251 μm (ideally)	301 μm (ideally)	301 μm
Stack gravimetric energy density	338.2 Wh kg⁻¹	340.2 Wh kg⁻¹	349.1 Wh kg⁻¹

Stack volumetric energy density	1304.4 Wh l ⁻¹	1057.1 Wh l ⁻¹	1084.7 Wh l ⁻¹
80% capacity retention cycle #	4 cycles	4 cycles	50 cycles

Supplementary Table 3. Estimated gravimetric and volumetric energy densities of AF-LMB cells with different current collector.

Supplementary Figure 21. Gravimetric energy density and cycle life of AF-LMB cells. a) Schematic illustration of a stacked AF-LMB. **b)** Estimated gravimetric energy densities of the AF-LMB with different current collectors. **c)** Number of cycles at 80% capacity retention for the AF-LMBs with different current collectors.

The gravimetric energy densities of a stacked AF-LMB with different current collector were calculated based on 0.1 C charge/discharge voltage profile (Fig. 7a). In this calculation, the single bi-cell constitutes one sheet of double-side coated NCM cathode, one sheet of fully charged anode (current collector + deposited Li), two sheet of separator and electrolyte²⁴. Here, in the case of Cu or p-CP, the thickness of the negative electrode increases significantly as Li deposition, but in the case of Cu, it is assumed that 5 μm per 1 mAh cm⁻², and in the case of p-CP, lithium is well deposited into the pore.

Supplementary Fig. 22. a-c, SEM images fully discharged state of anode current collectors after 10 cycles of 5 mAh cm⁻² Li plating/stripping at 2 mA cm⁻² for Cu (a), p-CP (b) and d-CP (c). Electrolyte: 1 M LiPF₆ EC/DEC 1:1 v/v 10% FEC 1% VC

In the case of the Cu current collector, it can be seen that a porous layer of about 100 μm is formed in only 10 cycles. When a bi-cell structured cell is constructed, the thickness of the single stack will be increased by 200 μm (100 μm each side), resulting in almost twice the volumetric energy density reduction (thickness vary from 251 at μm fully charged state to 413 μm at fully discharged state after 10 cycles), resulting volumetric energy density of 790.8 Wh l⁻¹. On the other hand, in the case of d-CP, it was confirmed that the thickness of the current collector was almost constant even after cycling.

9. Fig 6a presents coulombic efficiency on a scale from -10 to 110%. This is an unacceptable scale when CEs higher than 95% are required to even approach viability, and differences of just 1% make a very significant difference to lifetime. The minimum CE on this scale should be 90%.

Response: We appreciate the reviewer's comment. We agree with the reviewer that it is difficult to identify a CE difference of 1% on the current CEs scale. However, in the case of Cu and p-CP, since the CE is close to 90%, the data may be obscured, so the y-axis scale is set to 60%-105%, and then the average CE for each sample is indicated through the dotted line. As the reviewer suggested, Figure 6 was revised.

10. Throughout the paper, the authors claim their results have been achieved in an unpressurized system. On line 338 they state:

“It should be stressed that the cycle performance of d-CP in the AF-LMB is not derived from the pressure effect because stack pressure is not delivered to the inner surface of the 3D current collector.”

That may be so. However, it is my understanding that these results were achieved in a coin cell format. Unless the coin cells were constructed in some unconventional way which should be reported, then there is significant pressure exerted on the electrodes which will improve Li metal reversibility. Even if the authors above argument is true, there will be pressure exerted on the outer surfaces of the porous current collectors and Li plating there. Regardless, if the authors want to make the claim of achieving results in an unpressurized cell, they must present results that were obtained in a truly unpressurized cell (an unconstrained pouch, for example). The authors need to explain if these results are indeed from a conventional coin cell that has increased uniaxial pressure, and if so, remove or highly temper the claims of “unpressurized” results.

Response: We appreciate the reviewer for the comment. As in the reviewer's comment, pressure can be exerted on the electrode in the coin cell and it is not clear whether the pressure is delivered to the inner surface of 3D-current collector. To avoid confusion, so we removed the term ‘unpressurized’ and the related claims in the text.

Revised manuscript:

(Line 29-32)

While the efficacy of three dimensional (3D) porous current collectors in reducing the effective current density has been proven, their enlarged electroactive surface area can accompany a larger degree of electrolyte decomposition at the current collector-electrolyte interface¹².

(Line 57-60)

We report a successful introduction of 3D current collector in an NCM-based anode-free Li metal battery, that shows capacity retention of 90% at 50 cycles at high current density (2.0 mA cm⁻²), high areal capacity (4.2 mAh cm⁻² of Li_xNi_{0.8}Mn_{0.1}Co_{0.1}), and a lean carbonate electrolyte condition (E/C ratio of 4.0 μl mAh⁻¹).

(Line 409-411)

The introduction of NW defect significantly improved the cycling stability of an AF-LMB with high areal capacity (4.2 mAh cm^{-2}) under a lean electrolyte condition (E/C of $4.0 \mu\text{l mAh}^{-1}$) operating 50 cycles with 90 % capacity retention.

11. Line 58: “The defect-free SEI and Li-C orbital hybridization enable lateral Li deposition even in the internal structure of the 3D current collector where no external pressure is exerted normal on its surfaces”

I do not believe there is sufficient evidence to back up this claim. The authors provide top-down SEM images, but nothing that shows the morphology of the lithium in the internal porosity of the current collectors. The cross-sectional images seem to demonstrate that lithium indeed plates within the internal porosity, but it does not evidence that the lithium deposition inside the current collector is lateral. The authors seem to be inferring lateral deposition in the internal porosity from the lateral deposition demonstrated in Fig 5d-f which appears to be on the top surface of the current collector (which would be exposed to external pressure from the coin cell construction). I do not believe the authors have provided sufficient evidence to claim that they have enabled lateral Li deposition in the internal structure of the 3D current collector.

Response: Thanks for the critical comments from the reviewers. We tried to express the behavior of the growth of lithium nuclei on the carbon fiber with the word 'lateral'. This horizontal growth on the surface of the carbon fiber was observed at the initial nucleus growth or small Li deposition capacity about 3.0 mAh cm^{-2} , and then it was observed that the Li grows filling the pore between the fibers, and this is expressed as line 327-330 'By sharp contrast, for d-CP, its carbon fibers were uniformly covered by plated Li at 3 mAh cm^{-2} due to the uniform Li nucleation and lateral growth driven by the NW structure (Fig. 6f). Further deposition led to connection of the Li deposits to those on adjacent fibers (Fig. 6g, h).' We agreed that the expressions at introduction and conclusion section is ambiguous; therefore, it was corrected as 'initial lateral growth of Li nucleus and the resulting dense morphology'.

Revised manuscript:

(Line 6-9)

The nano-window defect-enriched surface induces a defect-free solid-electrolyte interphase by alleviating electron tunneling at the interface, and leads to a uniform Li nucleation and lateral growth of Li nucleus via Li-carbon orbital hybridization, resulting in a dense Li morphology.

(Line 50-53)

the NW structure addresses the critical issues at the anode interface by 1) constructing a thin and uniform SEI with reduced electrolyte decomposition and 2) evenly distributed and highly populated Li nucleus and its lateral growth on the current collector surface.

12. Fig 7d shows the lean electrolyte condition cycling up to 50 cycles. The d-CP show good capacity retention over this range, only 10% loss over 50 cycles (as reported in the abstract). However, Fig S16 shows that there is dramatic capacity roll over by cycle 60. I believe it is disingenuous for the full 100 cycle results to be hidden in the supplementary information. Fig 7c shows the flooded cells up to 100 cycles; the lean electrolyte cells should be shown to 100 cycles in the main paper as well. The challenges of lean electrolyte should not be minimized. The authors should be upfront about this and discuss these challenges in the main text.

Response: We appreciate the reviewer's comment. As the reviewer suggested, Figure 7 was modified. In addition, the current problems of the lean electrolyte cell are described in the main text.

Revised manuscript:

Fig 7: Electrochemical performance of AF-LMB with d-CP under realistic conditions. a, Voltage profiles of the anode-free NCM811 full cell with Cu or d-CP during the charge at 0.1 C and sequential constant voltage charging at 4.3 V (cut-off current: 0.02 C) and discharge. **b,** Comparison of the areal capacity and current density ensuring stable operation for this work and previous works. **c,** The plots of charge and discharge capacity as a function of cycle number for the anode-free NCM811 full cells with 1 M LiPF₆ EC/DEC 10% FEC 1% VC in a flooded cell condition (E/C ratio of 23.8 μl mAh⁻¹) and a lean electrolyte condition (E/C ratio of 4.0 μl mAh⁻¹). The flooded and lean electrolyte cells were cycled at a charge/discharge rate of 0.48 C (2 mA cm⁻²) with a potential window of 2.7-4.3 V.

(Line 364-367)

As displayed in Fig. 7b, the operation current density and the areal capacity of the AF-LMB with d-CP were higher than the previously reported values for liquid electrolyte based AF-LMBs^{2,38-47}, emphasizing the advantage of the NW structure.

(Line 384-385)

The rapid decrease in discharge capacity after 60 cycles is attributed to electrolyte depletion (Supplementary Fig. 20).

13. In their conclusion, the authors state that “a rational design strategy for advancing AF-LMBs is presented.” Can the authors comment about how practical large scale synthesis of these current collectors might be? Without a copper substrate, will these carbon materials be flexible enough to be used in practical format cells (wound pouches or cylinders)? Also, it is well known that lithium electrodes thicken during aging due to the accumulation of dead lithium. With these current collectors be strong enough to accommodate this expansion or might they crumble due to larger internal volume expansion?

Response: We are thankful for the reviewer's advice on improving the quality of this paper. The carbon paper can be bent to some extent, but it is not compatible with winding (bending radius ~5 cm). Therefore, stack-type pouch or prismatic cell are a possible format. For exploiting NW technology for jelly-roll based cells (cylindrical or prismatic cell), the NW defective carbons would be coated on flexible substrates such as polymers or metal foils by a wet coating method. For example, the carbonized ZIF-8 powders and binder are mixed in a solvent and the mixture is coated on a flexible substrate, followed by drying.

Regarding the issue of mechanical deterioration of carbon paper electrode, we agree with the reviewer's concern. In this work, Li deposit ($\leq 5 \text{ mAh cm}^{-2}$) did not completely fill the interstitial pores of d-CP, and thus stress would not be significant. However, at higher deposition capacity can cause the fracture of carbon fiber material. The conclusion "a rational design strategy for advancing AF-LMBs is presented" was made to deliver the efficacy of the concept of low Fermi-level surface. However, we recognized that it can be misinterpreted as the use of carbon fiber is advantageous. Therefore, we revised the conclusion, focusing on the surface design of the anode current collector for advancing AF-LMBs. Also, in the main text, we described the limitations of the carbon paper material used in our work, and the possibility of using the NW technology to modify the surfaces of other advanced carbon material substrates.

Revised manuscript:

(Line 395-398)

From a practical point of view, the use of carbon paper possesses a risk of mechanical deterioration at much larger Li plating capacity or at significant bending, hindering its use for winding-type cells. However, the NW technology could be used to modify the surfaces of various current collectors with higher mechanical strength and flexibility.

(Line 411-414)

Based on the model system provided herein regarding how the electronic structure of the current collector controls the SEI microstructure and Li nucleation, a rational design of surface electronic structure for advancing AF-LMBs is presented.

REVIEWERS' COMMENTS

Reviewer #1 (Remarks to the Author):

I am really impressed with the authors' thorough response to the comments I had raised. The new additions on the theory side make for a really great manuscript. I agree with the authors' assertion that lithium metal batteries should be distinguished from anode-free batteries.

V. Viswanathan